# Everything Everywhere All at Once: LLMs can In-Context Learn Multiple Tasks in Superposition

## Abstract

Large Language Models (LLMs) have demonstrated remarkable in-context learning (ICL) capabilities. In this study, we explore a surprising phenomenon related to ICL: LLMs can perform multiple, computationally distinct ICL tasks simultaneously, during a single inference call, a capability we term "task superposition". We provide empirical evidence of this phenomenon across various LLM families and scales and show that this phenomenon emerges even if we train the model to in-context learn one task at a time. We offer theoretical explanations that this capability is well within the expressive power of transformers. We also explore how LLMs internally compose task vectors during superposition. Furthermore, we show that larger models can solve more ICL tasks in parallel, and better calibrate their output distribution. Our findings offer insights into the latent capabilities of LLMs, further substantiate the perspective of "LLMs as superposition of simulators", and raise questions about the mechanisms enabling simultaneous task execution.

## 1 Introduction

Large Language Models (LLMs) have demonstrated remarkable capabilities across various domains, with one of the most intriguing being in-context learning (ICL) (Brown et al., 2020; Xie et al., 2022). ICL enables LLMs to perform tasks during inference without the need to fine-tune for that particular task, simply by providing a few examples within the input prompt. This ability has sparked significant interest in the research community, as it suggests that LLMs can adapt to novel tasks on-the-fly, using the capabilities that they acquired during pretraining, and the context provided.

While ICL has been extensively studied from both theoretical and empirical perspectives (Xie et al., 2022; Agarwal et al., 2024), many aspects of its underlying mechanisms remain elusive. In this work, we study a surprising phenomenon related to ICL that, to the best of our knowledge, has not been thoroughly studied before: LLMs can perform multiple distinct ICL tasks simultaneously, in a single inference call, a capability we refer to as *"task superposition"*.

Our study suggests that pretrained autoregressive LLMs such as Llama (Touvron et al., 2023) or GPT-3.5[1] (Brown et al., 2020) display superposition of tasks purely *in-context*[2]. When presented with multiple in-context examples from different tasks, in the same prompt, the models can generate outputs that correspond to solutions for all these individual tasks. For instance, given examples of addition and translation, the model can concurrently produce correct answers for both tasks, as well as the composition of these tasks (e.g., the result of addition translated into another language).

Figure 1 illustrates this phenomenon. In Figure 1a (left), given in-context examples of addition in different languages and the query "$91 + 83 \rightarrow$", the model generates probabilities for the correct sum in various languages, demonstrating its ability to perform addition and translation concurrently.

This discovery aligns and lends further support to the view of LLMs as superposition of simulators (Janus, 2022; Shanahan et al., 2023; Nardo, 2023) and the Bayesian perspective of ICL proposed by Xie et al. (2022). While not a mathematically rigorous formulation, we can conceptualize the output

---

[1]In particular, `gpt-3.5-turbo-instruct`.

[2]For other definitions of superposition, please see Section 2.

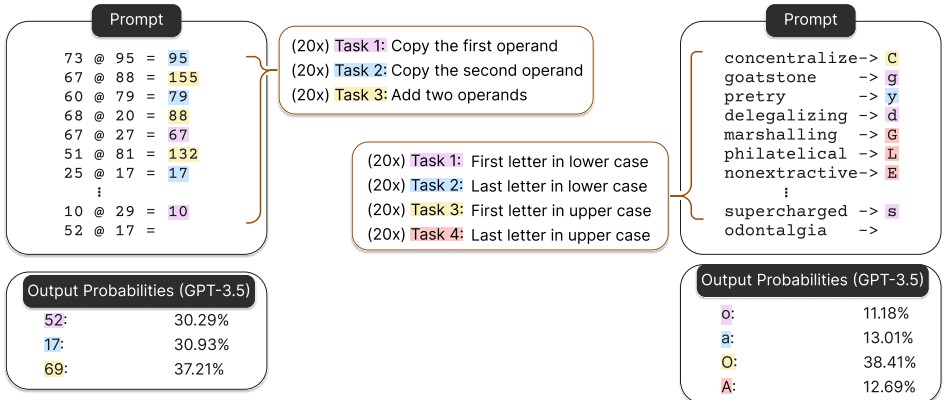

(a) **(left)** Two-digit addition in a variety of languages. **(right)** Naming the capital of a given country name, naming the continent of a given country name or capitalizing the country name.

(b) **(left)** Tasks `copy(op1)`, `copy(op2)` and `op1+op2`. **(right)** First or last letter in upper or lower case.

Figure 1: LLMs can perform task superposition. **(a)** Llama-3 70B and **(b)** GPT-3.5 Turbo are each presented with two sets of tasks. For each set of tasks, we show an example prompt such that all except the last row are in-context examples of one of the tasks and the last row is the query. We provide 20 in-context task examples for each task in the prompt with order randomized and provide the probabilities of outputs when correctly performing each task on the query.

of an LLM as a weighted sum of conditional probabilities across possible tasks:

$$\mathbb{P}(\text{output}|\text{prompt}) \approx \sum_{\text{task}} \mathbb{P}(\text{output}|\text{task}, \text{prompt})\mathbb{P}(\text{task}|\text{prompt}).$$

In this conceptual model, $\mathbb{P}(\text{output}|\text{prompt})$ represents the probability distribution over possible outputs given the input prompt, a task can be thought of as a latent variable representing different capabilities the model might possess (e.g., arithmetic, translation, sentiment analysis), $\mathbb{P}(\text{output}|\text{task}, \text{prompt})$ represents the output probability distribution if the model was specifically attempting to solve a single task, based on the test example in the prompt, and $\mathbb{P}(\text{task}|\text{prompt})$ represents the model's inferred probability that the prompt specifies a particular task.

While this mental model is a simplification of how an LLM operates, it provides an intuitive way to support the task superposition phenomenon we observe. Our findings lend support to the idea that LLMs can simultaneously maintain and utilize multiple task distributions, resulting in outputs that reflect a combination of relevant tasks.

**Our Contributions:**  Our study makes several key contributions:

1. Through extensive empirical investigation and theoretical results, we demonstrate that task superposition is prevalent across various pretrained LLM families (GPT3.5, LLama-3, Qwen).

2. We empirically show that task superposition emerges as we train on one task at a time.

3. We provide a theoretical construction showing that Transformers models are indeed capable of task superposition, and have the capacity to implement multiple tasks in parallel.

4. We explore how LLMs internally compose task vectors (Hendel et al., 2023) during superposition, and show how convex combinations of task vectors can reproduce the superposition effect.

5. We show that larger models can solve more tasks in parallel and more accurately reflect the distribution of in-context tasks.

We believe that our findings offer new insights into the latent capabilities of LLMs and raise questions about the mechanisms enabling simultaneous task execution. We believe this work sheds more light on the ICL capabilities of frontier language models, and offers a glimpse on potential applications of task superposition in practical settings.

## 2 RELATED WORK

**Theory and practice of in-context learning.** There is rich literature which formalizes in-context learning under diverse definitions. For example, prior works study in-context learning through a Bayesian framework for task retrieval (Xie et al., 2022; Panwar et al., 2023; Zhang et al., 2023), martingales (Falck et al., 2024), optimizers (Akyürek et al., 2023; Oswald et al., 2023; Dai et al., 2022) and more (Reddy, 2024; Olsson et al., 2022). Other works confirm the theoretical framing of in-context learning by using it to implement a variety of algorithms and methods (Zhou et al., 2023; Ahn et al., 2023; Giannou et al., 2023; Wu et al., 2024; Laskin et al., 2022; Zhou et al., 2022), or to approximate general-purpose computing machines (Giannou et al., 2023; Wei et al., 2022).

To bridge the gap between theory and practice, many works have used these theoretical insights to study in-context learning behaviors, such as in many-shot in-context learning, (Agarwal et al., 2024), long-context (Li et al., 2024), or eliciting personas (Choi & Li, 2024). Other works study the factors that influence how well models can learn through context, such as task diversity (Raventos et al., 2023; Chan et al., 2022), the balance between pre-training priors and in-context (Wei et al., 2023; Lin & Lee, 2024), in-context labels (Min et al., 2022; Lyu et al., 2022), and the in-context format (Lampinen et al., 2022). In-context learning has also been proposed as a means of fine-tuning to improve non-language tasks (Dinh et al., 2022).

The development of new architectures such as state space models (Gu & Dao, 2023) has further motivated studying whether in-context learning is prevalent in alternative architectures such as Mamba (Park et al., 2024; Grazzi et al., 2024; Zeng et al., 2024) or in looped transformers (Yang et al., 2023).

Steering models through in-context learning has been a growing area of interest. Recent work has hypothesized that in-context learning can be encapsulated by a high-dimensional description of a task, which can be used to replace, (Hendel et al., 2023) compose (Todd et al., 2024) or augment (Liu et al., 2024) the latent states of a model, in order to alter its default behavior. Task vectors can be combined via arithmetic operations to solve a variety of tasks (Ilharco et al., 2023). Prior work has also been investigating the power of tokens in defining a task (Bai et al., 2024).

**Other definitions of superposition.** Our findings on superposition are inspired by notions of language models as multiverse generators (Reynolds & McDonell, 2021; moire, 2021). One consequence of LLMs as a superposition of tasks is that the outputs may collapse to unintended simulacra, a behavior known as the "Waluigi effect" (Nardo, 2023).

Superposition has been defined in various related contexts of learning models. Feature superposition (Elhage et al., 2022) refers to a neural network's ability to represent multiple learned concept in a single neuron. Though our discovery of task superposition describes the same abstract idea, we stress that it is distinct from feature superposition because task superposition is most apparent in the final output of a model. Feature superposition is a microscopic-level observation whereas task superposition is a macroscopic-level observation.

Superposition is also described as a way to store multiple models in a single set of parameters (Cheung et al., 2019), processing multiple inputs simultaneously (Shen et al., 2024a; Murahari et al., 2022). In our work, we demonstrate task superposition directly as a result of language pre-training, without the necessity of additional adapters or decoding strategies.

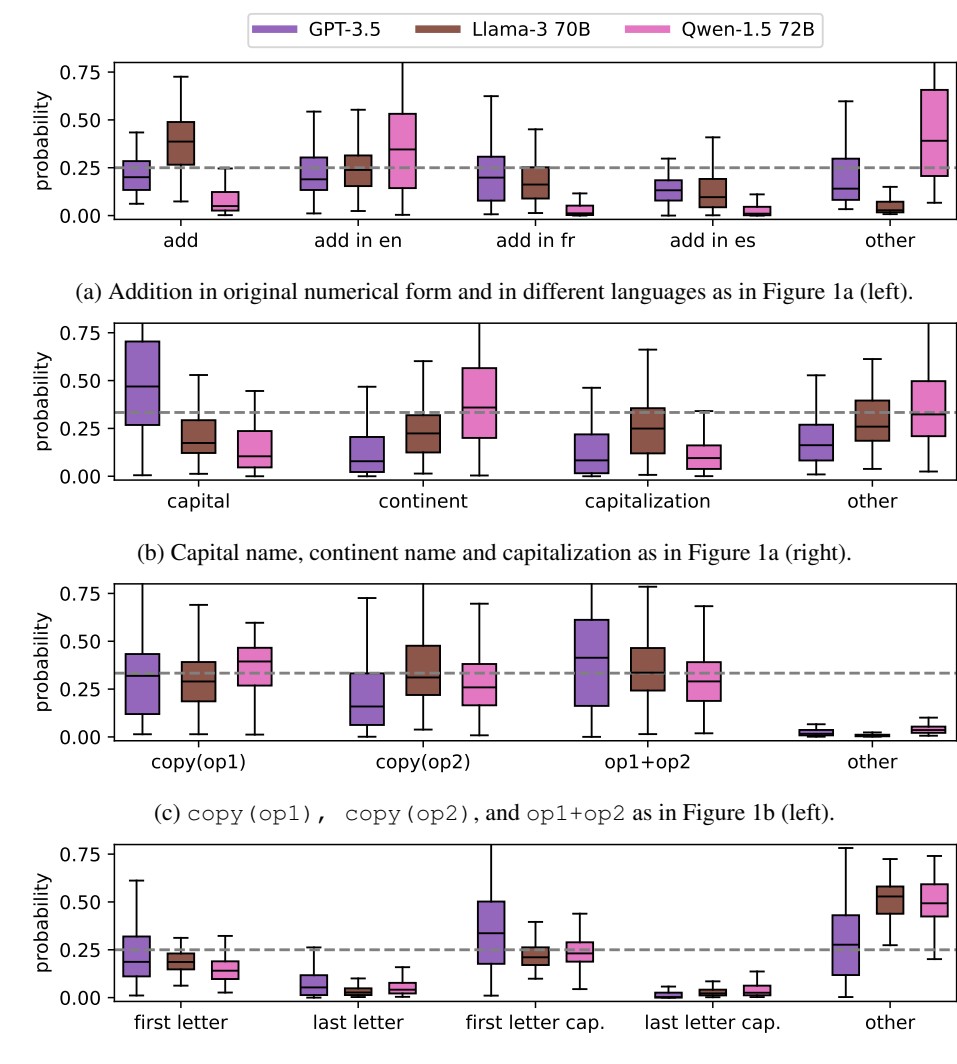

(a) Addition in original numerical form and in different languages as in Figure 1a (left).

(b) Capital name, continent name and capitalization as in Figure 1a (right).

(c) `copy(op1)`, `copy(op2)`, and `op1+op2` as in Figure 1b (left).

(d) First or last letter in upper or lower cases as in Figure 1b (right).

Figure 2: Distributions (0/25/50/75/100-percentiles) of probabilities for correct outputs of each task. For every set of tasks, we tested with 100 prompts and for each prompt, every task has 20 random in-context task examples with order randomized like in Figure 1. Category `other` is the sum of probabilities of all other outputs. Gray dashed line in each figure is the ideal probability if we assume the model perfectly calibrates its output distribution to the distribution of in-context task examples. With uniform distribution of task examples, the dashed lines are at 0.25 (4 tasks setting) and 0.33 (3 tasks setting).

## 3 LLMs ARE A SUPERPOSITION OF MULTIPLE IN-CONTEXT LEARNERS

In this section, we want to investigate if existing pre-trained models exhibit superposition of multiple tasks and whether this phenomenon is common (i.e., whether we can observe this phenomenon on a variety set of tasks and different families of LLMs).

> **Finding 1:** *LLMs can in-context learn multiple tasks in superposition when provided with prompts of a mixture of task examples.*

We denote $K$ by the number of tasks and consider four different settings of task mixtures.

1. Numerical addition and addition in English, French or Spanish ($K = 4$). Example prompt is shown in Figure 1a (left).

2. Given a name of a country, name the capital, continent or capitalize the country name ($K = 3$). Example prompt is shown in 1a (right).

3. Given input "{op1}@{op2}", copy op1, op2 or add op1 and op2 ($K = 3$). Example prompt is shown in Figure 1b (left).

4. Given a word, output first letter or last letter in lower or upper cases ($K = 4$). Example prompt is shown in 1b (right).

We provide GPT-3.5 (Brown et al., 2020), Llama-3 70B (AI@Meta, 2024) and Qwen-1.5 72B (Bai et al., 2023a) with prompts of uniform mixture of tasks (each task has 20 examples in the prompt ordered randomly). For each prompt consisting of in-context task examples (e.g., "$11 + 26 \rightarrow 37$" for the first task in the first setting) and a query (e.g., "$91 + 83 \rightarrow$"), we calculate the probabilities of outputs when correctly performing each task on the query and plot the distribution of probabilities for each task in Figure 2. Details on calculating the probabilities is in Appendix B.

Figure 2 reveals that in all four sets of tasks, all models have non-negligible median values of probabilities for at least two tasks. This indicates that the models can in-context learn multiple tasks in superposition when provided with prompts of a mixture of task examples.

We can also observe that, even though every task in a prompt has an equal number of in-context examples (20 examples), LLMs do not calibrate their output distribution perfectly with the in-context task example distribution and they still have bias on what task to perform. For example, Figure 2a shows that Llama-3 70B prefers performing numerical addition over addition in other languages, Qwen-1.5 72B prefers addition in English while GPT-3.5 does not have a strong preference over a single task. On the other hand, in Figure 2b GPT-3.5 has a strong preference over the capital task.

Additionally, some tasks are "harder" than other tasks. For example, in Figure 2d, all models assign near-zero probability for task answers of last_letter and last_letter_cap. The category other has relatively high values, indicating a high noise when prompted with in-context examples of this setting. In contrast, in Figure 2c, category other has very small values, indicating that all models most of the time would correctly assign the output probabilities to the correct answers.

## 4    TASK SUPERPOSITION IN MODELS TRAINED FROM SCRATCH

In Section 3 we investigated task superposition in pre-trained LLMs at inference time. We further investigate how task superposition emerges in LLMs during training. Specifically, if we train the model to in-context learn one task at a time, can it perform task superposition when provided with prompts containing examples of multiple tasks?

To answer this question, we train a small GPT-2 model (6 heads, 6 layers, $\sim$14million parameters) (Radford et al., 2019) to learn a family of retrieval tasks. The input has the form "{ch1}{ch2}{ch3}{ch4}{ch5}{ch6}{ch7}{ch8}$\rightarrow$" where ch1, ..., ch8 are distinct single characters. We consider 8 retrieval tasks – ret1, ..., ret8 – where ret1 is to output ch1 and so on. The model is trained to in-context learn one task (retrieve one of {ch1, ..., ch8}) at a time in training. Namely, during training, the model is only provided with text data such that each prompt only contains in-context examples of a single randomly chosen task (and different prompts can correspond to different tasks).

Concretely, for each sample, we randomly select task $t \in \{\text{ret1}, ..., \text{ret8}\}$ and inputs $\boldsymbol{x}^{(1)}, ..., \boldsymbol{x}^{(m)}$, where each $\boldsymbol{x}^{(j)}$ is an eight-character long string. We then form the sequence $\boldsymbol{s} = [\boldsymbol{x}^{(1)}, \boldsymbol{g}_t(\boldsymbol{x}^{(1)}), ..., \boldsymbol{x}^{(m)}, \boldsymbol{g}_t(\boldsymbol{x}^{(m)})]$ where $\boldsymbol{g}_t(\boldsymbol{x}^{(j)})$ is the output of performing task $t$ on $\boldsymbol{x}^{(j)}$. We train the model $M_\theta$ parametrized by $\theta$ using ICL training. In particular, we minimize the following objective:

$$\min_\theta \mathbf{E}_{\boldsymbol{s}} \left( \frac{1}{m-1} \sum_{j=1}^{m-1} \text{CE}(M_\theta(\boldsymbol{s}_j \oplus \boldsymbol{x}^{(j+1)}), \boldsymbol{g}_t(\boldsymbol{x}^{(j+1)})) \right), \tag{1}$$

where $\boldsymbol{s}_j \oplus \boldsymbol{x}^{(j+1)} \equiv [\boldsymbol{x}^{(1)}, \boldsymbol{g}_t(\boldsymbol{x}^{(1)}), ..., \boldsymbol{x}^{(j)}, \boldsymbol{g}_t(\boldsymbol{x}^{(j)}), \boldsymbol{x}^{(j+1)}]$ and CE is the cross-entropy loss.

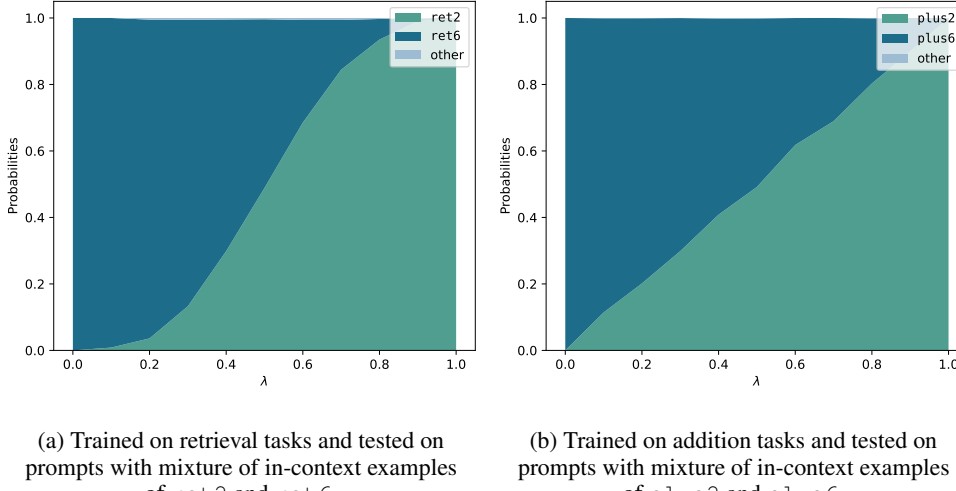

(a) Trained on retrieval tasks and tested on prompts with mixture of in-context examples of `ret2` and `ret6`.

(b) Trained on addition tasks and tested on prompts with mixture of in-context examples of `plus2` and `plus6`.

Figure 3: We consider two different settings of tasks: (a) given an eight-character length string as input, consider `ret1`, ..., `ret8` where `ret1` is to retrieve the first character and so on; and (b) given a two-digit integer as an input, consider `plus0`, ..., `plus9` where `plus0` is to add 0 on the input and so on. After training, for each setting, we select two tasks ad we provide the model with prompts containing in-context examples from these two tasks and vary the mixture ratio $\lambda$ such that the in-context task example distribution for two tasks is $[\lambda, 1 - \lambda]$. We plot $\lambda$ on x-axis and the output probabilities of task answers for each task on y-axis.

After training, we provide the model with prompts containing in-context examples of two tasks (in particular, we choose `ret2` and `ret6`) and see if the model performs task superposition. We vary the proportion of in-context examples of two tasks and plot the output distributions in Figure 3a.

Similarly, we consider a second setting involving 10 tasks. Given a two digit integer input `num`, task `plus0` outputs `num`, task `plus1` outputs $num + 1$ and so on, up to task `plus9`. The model is trained to in-context learn one of `plus0`,..., `plus9` at a time, following the procedure above. During inference time, the model is tested with prompts containing a mixture of in-context examples from tasks `plus2` and `plus6`. We vary the mixture ratio and show the output distributions in Figure 3b.

> **Finding 2:** *Transformers can in-context learn multiple tasks in superposition even if trained to in-context learn one task at a time.*

Remarkably, from Figure 3a and 3b, GPT-2 trained from scratch to in-context learn one task at a time can generalize to simultaneously performing multiple tasks and calibrate the output probabilities according to the in-context task example distribution when provided with a mixture of in-context examples. For example, in Figure 3a at the mixture ratio $\lambda = 0.5$, meaning that 50 percent of the examples in the prompt is from task `ret2` and the other $50\%$ comes from task `ret6`, we can see the output probabilities for task answers of `ret2` and `ret6` being roughly $[0.5, 0.5]$. We can observe similar behavior in Figure 3b.

## 5 TRANSFORMERS HAVE THE CAPACITY TO PERFORM TASK SUPERPOSITION

In this section, we explore whether Transformers have the inherent expressivity to perform multiple tasks in superposition with a single inference call. To this end, we provide a theoretical construction of a Transformer which, given the ability to implement multiple tasks, performs task superposition depending on the examples given in-context.

**Theorem 1.** *A seven layer transformer with embedding dimension $\mathcal{O}(d + \log(mn))$ with $K$ heads per attention layer can perform $K$ tasks on vectors of dimension $d$ in superposition, with weighting based on $m$ different in-context examples each of length $n$.*

The proof of Theorem 1 is provided in Appendix D.4. Note that while this does not guarantee that training a Transformer will actually find these parameters, it does indicate that Transformers are expressive enough to perform task superposition at test time. Below we outline the main ideas used in the proof.

**Prediction based on multiple tasks.** Assume that we are given $m$ in-context samples $(\boldsymbol{x}_1^{(j)}, \ldots, \boldsymbol{x}_{n-2}^{(j)}, \text{'='}, \boldsymbol{y}^{(j)})_{j=1}^m$ where '=' represents a specific value used only for preceding the label, and a set of $k$ different Transformers $\text{TF}_i$ which can implement the $T$ different desired tasks, where each deterministic task is denoted as $\boldsymbol{g}_i(\boldsymbol{x}^{(j)})$ with $i \in [k]$ and $j \in [m]$, *i.e.* $\boldsymbol{y}^{(j)} = \boldsymbol{g}_i(\boldsymbol{x}^{(j)})$ for some task $i$ dependent on sample $j$. Using the weights of each $\text{TF}_i$, we can compute outputs of the following form:

$$
\begin{bmatrix}
\cdots & \boldsymbol{x}_1^{(j)} & \cdots & \boldsymbol{x}_{n-2}^{(j)} & = & \boldsymbol{y}^{(j)} & \cdots \\
\cdots & \mathbf{0} & \cdots & \mathbf{0} & \mathbf{0} & \mathbf{0} & \cdots \\
& \vdots & & \vdots & \vdots & \vdots & \\
\cdots & \mathbf{0} & \cdots & \mathbf{0} & \mathbf{0} & \mathbf{0} & \cdots
\end{bmatrix}
\rightarrow
\begin{bmatrix}
\cdots & \boldsymbol{x}_1^{(j)} & \cdots & \boldsymbol{x}_{n-2}^{(j)} & = & \boldsymbol{y}^{(j)} & \cdots \\
\cdots & \mathbf{0} & \cdots & \mathbf{0} & \mathbf{0} & \|\boldsymbol{g}_1(\boldsymbol{x}^{(j)}) - \boldsymbol{y}^{(j)}\|_1 & \cdots \\
& \vdots & & \vdots & \vdots & \vdots & \\
\cdots & \mathbf{0} & \cdots & \mathbf{0} & \mathbf{0} & \|\boldsymbol{g}_T(\boldsymbol{x}^{(j)}) - \boldsymbol{y}^{(j)}\|_1 & \cdots
\end{bmatrix}
$$

We use the $l_1$ norm to aggregate the prediction, in case that the task is multi-dimensional. These differences are used to identify tasks, as $\|\boldsymbol{g}_i(\boldsymbol{x}^{(j)}) - \boldsymbol{y}^{(j)}\|_1 \approx 0$ for $\boldsymbol{y}^{(j)}$ coming from task $i$. Different heads at each layer in the model are used to execute each of the tasks in parallel using the weights from $\text{TF}_i$. In Appendix D we construct tasks where an arbitrary function $\boldsymbol{g}_i(\boldsymbol{x}_l^{(j)})$ is implemented using ReLUs for some fixed $l$ that is task-specific.

**Creating task identifiers.** Having the differences between the implemented function and the label, we first use the ReLUs to clean up the vectors $\boldsymbol{v}_k$ so that only the positions in each vector that are associated with a task are maintained and the rest are set to $1$[3]. We thus create the vectors $(\boldsymbol{v}_k')_i = \|\boldsymbol{g}_k(\boldsymbol{x}_{1:l-1}^{(j)}) - \boldsymbol{x}_l^{(j)}\|_1$ and $(\boldsymbol{v}')_* = 1$ otherwise. Now we use ReLUs to threshold and create an indicator vectors $\mathbf{1}_{\{\|\boldsymbol{g}_k(\boldsymbol{x}_{1:l-1}^{(j)}) - \boldsymbol{x}_l^{(j)}\|_1 \approx 0\}}$ which identify the task, *i.e.*,these are task identifiers. Notice that if the task is correctly predicted then the difference should be close to $0$ (up to some error), while if the task is not identified the corresponding value would not be $0$; the rest of the rows would be $1$. We have created one vector for each task, which has $1$ in the position of the corresponding task if the task was identified in the context.

**Averaging and task superposition.** As a last step, we average all the task identifiers and place the result in the last column, in which the next prediction will happen. We then use the averaged task identifier to weight the prediction of each task based on it, as in task superposition. If the task has been identified multiple times in the context, it would be assigned a higher weight/probability.

# 6 TASK SUPERPOSITION THROUGH THE LENS OF TASK VECTORS

While in Section 5 we provide an existential result by constructing a Transformer that performs task superposition and shows that task superposition is well within the expressive power of Transformers, we would like to further investigate how task superposition manifest in pretrained LLMs internally. In this section we explore the underlying mechanisms that LLMs employ during task superposition. In particular, we focus our empirical study on *task vectors* (Hendel et al., 2023) where the detailed implementation is in Appendix C. Task vectors are vectors in the embedding space and are found to encode the algorithm that a model internally implements to solve a task given in-context demonstrations.

We want to investigate if there is any relation between the task vectors of each individual task and the task vectors of a mixture of task examples in the prompt. To this end, we consider two sets of tasks:

(a) `copy(op1)`, `copy(op2)` and `op1+op2` as in Figure 1b (left).

(b) Given a two-digit integer, task `to_fr` translates it to French, task `to_de` translates it to German and task `to_it` translates it to Italian.

---

[3]This step is not mandatory, but it ensures that we have no values over which we have no control. We leave as future work an error analysis on how these values could affect the task identifiers.

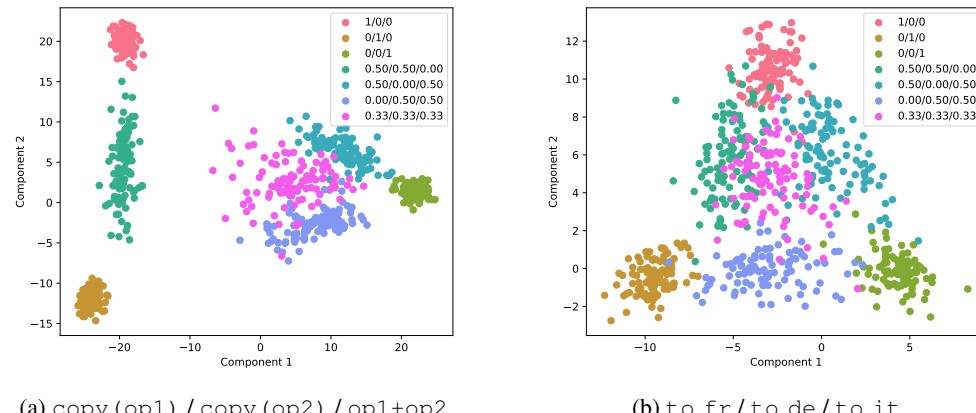

(a) `copy(op1)` / `copy(op2)` / `op1+op2`

(b) `to_fr` / `to_de` / `to_it`

Figure 4: Task vectors of Llama-3 8B projected onto two axes chosen by LDA for two sets of tasks: **(a)** `copy(op1)`, `copy(op2)` and `op1+op2` and **(b)** `to_fr`, `to_de` and `to_it`. For tasks $t_1, t_2, t_3$, we use "$\mathbb{P}(t_1)/\mathbb{P}(t_2)/\mathbb{P}(t_3)$" to denote different levels of task mixtures, e.g., "0.50/0.50/0.00" represents the case where the in-context task examples are 50% $t_1$, 50% $t_2$ and 0% $t_3$.

For each set of tasks, we collect the task vectors for each individual task and task vectors extracted from prompts that contain examples of different tasks. In Figure 4, we project task vectors along two axes chosen by linear discriminant analysis (LDA).

> **Finding 3:**   *LLMs internally combine task vectors during task superposition.*

Interestingly, we observe that the locations of task vectors of a mixture of tasks strongly correlate with the locations of task vectors for each individual task and the in-context task example distribution (the mixture ratio for examples of different tasks). For example, if the prompt includes an equal number of in-context examples from each task, the task vectors are roughly centered in the middle; if the prompt only contains in-context examples of two tasks, then the task vectors roughly lie on the connecting line between task vectors of two individual tasks. We argue that this observation is indicative of the fact that, when prompted with a mixture of in-context task examples, LLMs internally combine task vectors.

As we observe signs that LLMs internally compose task vectors, we want to further investigate whether we can reproduce the task superposition phenomenon by patching in a convex combination of task vectors. For example, for tasks `copy(op1)` and `copy(op2)`, we first extract the corresponding task vectors $V_{copy(op1)}$ and $V_{copy(op2)}$ on Llama-3 8B using the method described in Appendix C. We then make a convex combination of the two task vectors with parameter $\lambda$ that controls the ratio:

$$V_{\text{interpolate},\lambda} = \lambda \cdot V_{copy(op1)} + (1 - \lambda) \cdot V_{copy(op2)}.$$

> **Finding 4:**   *Convex combinations of task vectors produce task superposition.*

For a new query (in this scenario in the form "$\{op1\}@\{op2\}=$"), we patch the vector $V_{\text{interpolate},\lambda}$ into the model at the task vector layer. We calculate the model output probabilities that correspond to each task while we vary $\lambda$. For each $\lambda$, we use 100 different queries and plot the average probabilities in the top row of Figure 5. As a comparison, in the bottom rows of Figure 5, we plot the corresponding output probabilities when providing the models with prompts containing mixture of task examples where the mixture ratio is controlled by $\lambda$.

In top row of Figure 5, we observe that patching convex combinations of task vectors into the model produces task superposition. We would also like to point out that in Figure 5b, although irrelevant outputs sum up to a large probability, the task answers for two tasks `to_de` and `to_it` in most cases will still be the top-2 answers.

Comparing the top rows and the bottom rows, we can see that top rows (the scenario of interpolating task vectors of individual tasks) have larger probabilities of irrelevant output (category `other`).

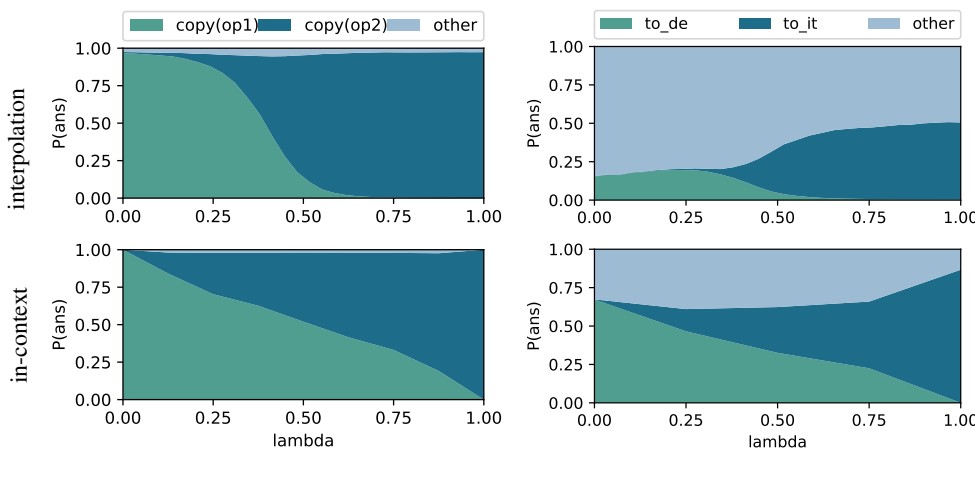

(a) Tasks: `copy(op1)` and `copy(op2)`   (b) Tasks: translate `to_de` and `to_it`

Figure 5: On Llama-3 8B, we vary the proportion, $\lambda$, between two tasks and observe how the output probabilities for the correct answers change. The proportion $\lambda$ is varied in two ways: (1) in the top row, we plot the output from patching in a convex combination of task vectors for two tasks. (2) in the bottom row, we plot the output from a mixed proportion of in-context examples for the two tasks. Subplot (a) shows the output probabilities from mixing two copy tasks and (b) shows the probabilities from mixing two translate tasks.

Task vector interpolation also produces less of a linear relationship between $\lambda$ and the output probabilities. This shows that while convex combinations of task vectors are sufficient for producing task superposition, this does not fully explain task superposition. We leave it to future work to investigate other mechanistic explanations of task superposition.

## 7   TASK SUPERPOSITION CAPABILITIES AS THE MODEL SCALES

**Finding 5:** *Within the same LLM family, bigger models can solve more tasks in parallel and better calibrate to ICL distribution.*

We want to further investigate how models' task superposition capabilities changes as the model size scales. In particular, we investigate two questions: 1) whether larger models can perform more tasks in-context and 2) whether larger models can align their output distribution more closely with the distribution of task examples provided in the prompt. We chose the Qwen-1.5 model family since it contains several model sizes ranging from 0.5B to 14B parameters.

We first introduce a quantity which captures the capability of a model to perform multiple tasks. Given a prompt that contains examples of $K$ tasks, we define $r$ to be the number of these tasks whose correct answers appear among the model's top-$K$ most likely outputs. Note that $r \leq K$.

To see how close the model align the output distribution with the distribution of task examples, we use KL-divergence defined below:

$$\text{KL}(\mathcal{P}||\mathcal{D}) = \sum_{x \in X} \mathcal{P}(x) \log \left( \frac{\mathcal{P}(x)}{\mathcal{D}(x)} \right), \tag{2}$$

where $\mathcal{P}$ is the models' probabilities on the outputs when correctly performing each task on the query and $\mathcal{D}$ is the in-context task example distribution. For example the prompt in Figure 1a (left) gives $\mathcal{P} = [0.5217, 0.1316, 0.1110, 0.2169, ...]$ and $\mathcal{D} = [0.25, 0.25, 0.25, 0.25, 0, ...]$.

We consider the setting of $K = 6$ different tasks: given an input of the form "{num}$\rightarrow$" where `num` is a two-digit integer, we consider 6 tasks that output (1) `num` itself, (2) negation of `num`, (3) `num` $+ 1$, (4) `num` $- 1$, (5) `num` $\times 2$ and (6) `num`$^2$.

We choose the number of in-context examples $m = 60$ (each task has 10 examples) and configure the prompt with three different in-context task example distributions $\mathcal{D}_1, \mathcal{D}_2$ and $\mathcal{D}_3$. In particular, $\mathcal{D}_1$ is the uniform distribution, $\mathcal{D}_2$ has probability 0.5 on the third task and 0.1 on other tasks, and $\mathcal{D}_3$ is a distribution with probabilities alternating between 0.25 and 0.083.

For each in-context task examples distribution $\mathcal{D}_i$, we generate 100 prompts and for each prompt we calculate the probabilities of outputs when correctly performing each task. The average values of $r$ and KL-divergence under three distributions are shown in Figure 6.

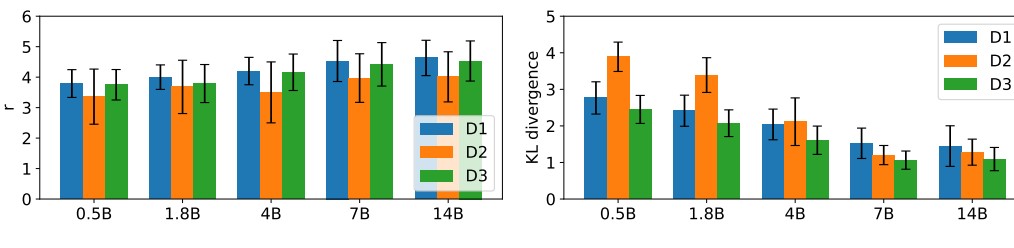

(a) $r$ (the number of tasks whose correct answers appear in top-$K$ most likely outputs).

(b) KL divergence.

Figure 6: (a) Average number of tasks completed, $r$, and (b) KL divergence for Qwen-1.5 model family under ICL distributions $\mathcal{D}_1, \mathcal{D}_2$ and $\mathcal{D}_3$ where $\mathcal{D}_1$ is the uniform distribution, $\mathcal{D}_2$ has probability 0.5 on the third task and 0.1 on other tasks, and $\mathcal{D}_3$ is a distribution with probabilities alternating between 0.25 and 0.083.

In Figure 6a, we can observe that bigger models have higher $r$ values (except for task distribution $\mathcal{D}_2$, 4B model has slightly lower $r$ than that of the 1.8B model). This shows bigger models will have more correct answers of tasks show up in their top-$K$ probable outputs and therefore they can solve more tasks at the same time. In Figure 6b, we can see that for larger models like Qwen-1.5 7B and Qwen-1.5 14B, the KL-divergence values are small, and for each model, the differences between KL-divergence values under in-context task example distributions $\mathcal{D}_1, \mathcal{D}_2$ and $\mathcal{D}_3$ are small. This indicates that bigger models can better calibrate their output distribution to the in-context task example distribution.

## 8    LIMITATIONS AND FUTURE DIRECTIONS

One limitation of our work is the current gap between the demonstrated capability of LLMs to perform task superposition and its practical application in real-world scenarios. While we have shown that LLMs possess the capacity to execute multiple tasks simultaneously, conventional decoding algorithms are not equipped to fully leverage this capability. This limitation stems from what we term "generation collapse," a phenomenon where, after the first token is generated, the model tends to converge on predicting tokens for a single task, effectively negating its ability for multi-task execution.

This collapse presents a substantial challenge in harnessing the full power of task superposition. It highlights a critical area for future research: developing decoding strategies that can maintain the model's multi-task state throughout the generation process. Recent work by Shen et al. (2024b) offers some hope that this direction may be fruitful, by proposing a "superposed decoding" algorithm. Their method efficiently generates multiple streams of tokens from a single inference pass by utilizing superposed token embeddings. While this approach represents a significant step forward, it also highlights the potential for further innovation in this area.

## 9    CONCLUSION

We report on the discovery of task superposition, which is the ability of LLMs to simultaneously solve distinct tasks from in-context examples. Task superposition is present in a variety of pretrained models, and becomes more accurate at predicting the distribution of tasks as the model size increases. We also find evidence that while displaying task superposition, models internally mix the task vectors of each individual task. We hope that our findings will contribute to understanding in-context learning mechanisms and enhance our knowledge of LLMs overall.

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

## A  NOTATIONS

| Notation | Description |
|----------|-------------|
| $K$ | Number of tasks |
| $l$ | Length of a task's output |
| $\ell$ | layer $\ell$ for a model |
| $m$ | Number of in-context examples |
| $n$ | Length of each in-context example |
| $\mathcal{V}$ | Token vocabulary |
| $\boldsymbol{g}_i(\cdot)$ | Operation performed by Task $i$ |
| $\boldsymbol{x}^{(j)}$ | Data for example $j$ |
| $\boldsymbol{y}^{(j)}$ | Label for example $j$ |
| $\boldsymbol{s}_m$ | $m$ in-context examples |
| $\boldsymbol{f}(\cdot)$ | Model (predictor) |
| $\boldsymbol{p}$ | Positional encodings |

## B   IMPLEMENTATION DETAILS ON CALCULATING PROBABILITIES

In this section we provide details on how we calculate probabilities of different outputs given a prompt in our setting.

**Notations.**   Let $\mathcal{V}$ be the token vocabulary, `LM` be an LLM, $T$ be the tokenizer. We use "..." to represent a string, $\langle...\rangle$ to represent a single token where the content within the angle brackets is an integer representing token's index in vocabulary. For example, token `<266>` corresponds to "at". We use [`<...>`, `...`, `<...>`] to represent a sequence of tokens. Given a tokenizer, we use two functions $\mathrm{tok}(\cdot)$ and $\mathrm{detok}(\cdot)$ to tokenize strings and detokenize tokens. For example $\mathrm{tok}(\text{"superposition"}) = [\texttt{<9712>},\texttt{<3571>}]$ and $\mathrm{detok}([\texttt{<16>},\texttt{<10>},\texttt{<16>},\texttt{<28>},\texttt{<17>}]) = $ "1+1=2".

In our in-context learning setting, an input string consists of in-context examples (separated by the delimiter "\n") and a query. For example, an example prompt can be "1+1=2\n2+2=4\n3+3=".

We view an LLM as a next-token predictor that outputs a probability distribution over the token space given input and there is a corresponding $\mathbb{P}(\cdot|\cdot)$ such that given a sequence of tokens $[v_1, ..., v_M]$ where $v_j \in \mathcal{V}$, $\mathbb{P}(u \mid [v_1, ..., v_M])$ measures the probability of the next token being $u$ where $u \in \mathcal{V}$.

**Measuring the probabilities of task answers.**   Let $I$ be the input prompt. For example, in the example in Figure 1a (left), the prompt is "11+26->37\n33+13->quarante-six\n ...30+25->fifty-five\n91+83->". We consider four tasks: 1) numerical addition, 2) addition in English, 3) addition in French and 4) addition in Spanish. The corresponding task answers (the output of correctly performing task on the query) are "174", "one hundred and seventy-four", "cent soixante-quatorze" and "ciento setenta y cuatro", respectively. We want to measure the probability of each task answer.

Let $o$ be a task answer in string. Let $[v_1, ..., v_M] := \mathrm{tok}(I)$ and let $[u_1, ..., u_N] := \mathrm{tok}(o)$. Then the probability of the task answer $o$ given prompt $I$ can be calculated as

$$\mathbb{P}(u_1 \mid [v_1, ..., v_M]) \prod_{j=2}^{N} \mathbb{P}(u_j \mid [v_1, ..., v_M, u_1, ..., u_{j-1}]). \tag{3}$$

## C   IMPLEMENTATION DETAILS ON TASK VECTORS

We use the task vector definition from Hendel et al. (2023). For example, for task `copy(op1)` in Figure 1b (left), the procedure to collect the task vector consists of

1. Collect a dataset of 100 ICL sample prompts. Each prompt consists of $m = 60$ in-context examples of a particular task and a query $\boldsymbol{x}^{(m+1)}$. Each task example $(\boldsymbol{x}^{(j)}, \boldsymbol{y}^{(j)})$ follows the form "{op1}@{op2}={op1}", where $\boldsymbol{x}^{(j)}$ has the form "{op1}@{op2}=" and $\boldsymbol{y}^{(j)}$ is performing task `copy(op1)` on $\boldsymbol{x}^{(j)}$, namely op1.

2. For each prompt $\equiv \boldsymbol{s} = [\boldsymbol{x}^{(1)}, \boldsymbol{y}^{(1)}, ..., \boldsymbol{x}^{(m)}, \boldsymbol{y}^{(m)}, \boldsymbol{x}^{(m+1)}]$ in the dataset, we feed $\boldsymbol{s}$ into the transformer model $\boldsymbol{f}$, and extract the feature (which is a vector) at the last "=" token in layer $\ell$. Call this vector $\boldsymbol{f}(\boldsymbol{s}; \ell)$. Then we average $\boldsymbol{f}(\boldsymbol{s}; \ell)$ across all prompt $\boldsymbol{s}$ to get $\boldsymbol{v}(\ell)$ for layer $\ell$.

3. Now for each layer $\ell$ we have a vector $\boldsymbol{v}(\ell)$. We run a forward pass with one query $\boldsymbol{x}$ in the form "{op1}@{op2}=" and we patch in $\boldsymbol{v}(\ell)$ at the "=" token position in layer $\ell$, simulating the effect of a complete context. We repeat this process 100 times for different query $\boldsymbol{x}$ and get an accuracy $acc_\ell$ of performing task `copy(op1)` with vector $\boldsymbol{v}(\ell)$.

4. The task vector layer $\ell^*$ is selected by

$$\ell^* = \arg\max_\ell acc_\ell,$$

and we define the task vector $\mathrm{V}_{\texttt{copy(op1)}} := \boldsymbol{v} = \boldsymbol{v}(\ell^*)$.

Here we record the task vector layer where task vectors are extracted in Section 6.

| Task | Task vector layer |
|:---:|:---:|
| `copy(op1)`, `copy(op2)`, `op1+op2` | 14 |
| `to_de(op1)`, `to_fr(op1)`, `to_it(op1)` | 19 |

Table 1: Task vector layer for various tasks considered in Section 6.

# D CONSTRUCTION DISPLAYING SUPERPOSITION

In this section we construct a Transformer that is performing superposition of multiple tasks at inference. For this purpose, we first construct a Transformer that copies from $n$-tuple in-context examples the $i$-th one, as well as any function using the ReLU layers. We then create indicator vectors, for each task, which show whether a specific task is present in-context or not. As a last step, we combine these indicator vectors to create the superposition of different tasks. Notice that using the parallel heads of the transformer architecture we can process each task independently until the last step in which the predictions are combined.

## D.1 OVERVIEW

Here we provide a brief overview of how the construction is implemented, while latter we provide the corresponding details.

**Prediction based on multiple tasks.** Assume that we are given $m$ in-context samples $(\boldsymbol{x}_1^{(j)}, \ldots, \boldsymbol{x}_{n-2}^{(j)}, \text{‘=’}, \boldsymbol{y}^{(j)})_{j=1}^m$ where ‘=’ represents a specific value used only for preceding the label, and a set of $k$ different Transformers $\text{TF}_i$ which can implement the $T$ different desired tasks, where each deterministic task is denoted as $\boldsymbol{g}_i(\boldsymbol{x}^{(j)})$ with $i \in [k]$ and $j \in [m]$, *i.e.* $\boldsymbol{y}^{(j)} = \boldsymbol{g}_i(\boldsymbol{x}^{(j)})$ for some task $i$ dependent on sample $j$. Using the weights of each $\text{TF}_i$, we can compute outputs of the following form:

$$
\begin{bmatrix}
\ldots & \boldsymbol{x}_1^{(j)} & \ldots & \boldsymbol{x}_{n-2}^{(j)} & \boldsymbol{=} & \boldsymbol{y}^{(j)} & \ldots \\
\ldots & \boldsymbol{0} & \ldots & \boldsymbol{0} & \boldsymbol{0} & \boldsymbol{0} & \ldots \\
 & \vdots & & \vdots & \vdots & \vdots & \\
\ldots & \boldsymbol{0} & \ldots & \boldsymbol{0} & \boldsymbol{0} & \boldsymbol{0} & \ldots
\end{bmatrix}
\rightarrow
\begin{bmatrix}
\ldots & \boldsymbol{x}_1^{(j)} & \ldots & \boldsymbol{x}_{n-2}^{(j)} & \boldsymbol{=} & \boldsymbol{y}^{(j)} & \ldots \\
\ldots & \boldsymbol{0} & \ldots & \boldsymbol{0} & \boldsymbol{0} & \|\boldsymbol{g}_1(\boldsymbol{x}^{(j)}) - \boldsymbol{y}^{(j)}\|_1 & \ldots \\
 & \vdots & & \vdots & \vdots & \vdots & \\
\ldots & \boldsymbol{0} & \ldots & \boldsymbol{0} & \boldsymbol{0} & \|\boldsymbol{g}_T(\boldsymbol{x}^{(j)}) - \boldsymbol{y}^{(j)}\|_1 & \ldots
\end{bmatrix}
$$

We use the $l_1$ norm to aggregate the prediction, in case that the task is multi-dimensional. These differences are used to identify tasks, as $\|\boldsymbol{g}_i(\boldsymbol{x}^{(j)}) - \boldsymbol{y}^{(j)}\|_1 \approx 0$ for $\boldsymbol{y}^{(j)}$ coming from task $i$. Different heads at each layer in the model are used to execute each of the tasks in parallel using the weights from $\text{TF}_i$. In Appendix D we construct tasks where an arbitrary function $\boldsymbol{g}_i(\boldsymbol{x}_l^{(j)})$ is implemented using ReLUs for some fixed $l$ that is task-specific.

**Creating task identifiers.** Having the differences between the implemented function and the label, we first use the ReLUs to clean up the vectors $\boldsymbol{v}_k$ so that only the positions in each vector that are associated with a task are maintained and the rest are set to $1$[4]. We thus create the vectors $(\boldsymbol{v}_k')_i = \|\boldsymbol{g}_k(\boldsymbol{x}_{1:l-1}^{(j)}) - \boldsymbol{x}_l^{(j)}\|_1$ and $(\boldsymbol{v}_*')_* = 1$ otherwise. Now we use ReLUs to threshold and create an indicator vectors $\mathbf{1}_{\{\|\boldsymbol{g}_k(\boldsymbol{x}_{1:l-1}^{(j)}) - \boldsymbol{x}_l^{(j)}\|_1 \approx 0\}}$ which identify the task, *i.e.* these are task identifiers. Notice that if the task is correctly predicted then the difference should be close to $0$ (up to some error), while if the task is not identified the corresponding value would not be $0$; the rest of the rows would be $1$. We have created one vector for each task, which has $1$ in the position of the corresponding task if the task was identified in the context.

**Averaging and task superposition.** As a last step, we average all the task identifiers and place the result in the last column, in which the next prediction will happen. We then use the averaged task identifier to weight the prediction of each task based on it, as in task superposition. If the task has been identified multiple times in the context, it would be assigned a higher weight/probability.

## D.2 TASK IDENTIFICATION

The first task for performing task superposition based on in-context examples is to define a set of tasks that the model is able to implement.

First, the outputs of tasks need to be identified.

---

[4]This step is not mandatory, but it ensures that we have no values over which we have no control. We leave as future work an error analysis on how these values could affect the task identifiers

**Lemma 1.** *Consider the following input*

$$X = \begin{bmatrix} x_1^{(1)} & \dots & y^{(j-1)} & x_1^{(j)} & \dots & x_{n-2}^{(j)} & = & y^{(j)} & x_1^{(j+1)} & \dots \\ 0 & \dots & 0 & 0 & \dots & 0 & 1 & 0 & 0 & \dots \\ \mathbf{0} & \dots & \mathbf{0} & \mathbf{0} & \dots & \mathbf{0} & \mathbf{0} & \mathbf{0} & \mathbf{0} & \dots \end{bmatrix},$$

*where $x_i^{(j)} \in \mathbb{R}^{d-1}$ before the positional encodings are added, with one additional dimension that represents if the symbol is an 'equals' symbol. Then, a 1-layer transformer with a single attention head and embedding dimension $\mathcal{O}(d + \log(mn))$ can output*

$$X = \begin{bmatrix} x_1^{(1)} & \dots & y^{(j-1)} & x_1^{(j)} & \dots & x_{n-2}^{(j)} & = & y^{(j)} & x_1^{(j+1)} & \dots \\ 0 & \dots & 1 & 0 & \dots & 0 & 0 & 1 & 0 & \dots \end{bmatrix}$$

*Proof.* With positional encodings appended, let the input have the following structure:

$$X = \begin{bmatrix} x_1^{(1)} & \dots & x_1^{(j)} & x_2^{(j)} & \dots & x_{n-2}^{(j)} & = & y^{(j)} & x_1^{(j+1)} & \dots \\ 0 & \dots & 0 & 0 & \dots & 0 & 1 & 0 & 0 & \dots \\ \boldsymbol{p}_{n+1} & \dots & \boldsymbol{p}_{jn+1} & \boldsymbol{p}_{jn+2} & \dots & \boldsymbol{p}_{jn+n-2} & \boldsymbol{p}_{jn+n-1} & \boldsymbol{p}_{jn+n} & \boldsymbol{p}_{(j+1)n+1} & \dots \\ \boldsymbol{p}_n & \dots & \boldsymbol{p}_{jn} & \boldsymbol{p}_{jn+1} & \dots & \boldsymbol{p}_{jn+n-3} & \boldsymbol{p}_{jn+n-2} & \boldsymbol{p}_{jn+n-1} & \boldsymbol{p}_{(j+1)n} & \dots \end{bmatrix} \tag{4}$$

To rotate the second row one position to the right, use the following matrices.

$$\boldsymbol{W}_Q = \begin{bmatrix} \mathbf{0} & \mathbf{0} & \mathbf{0} & \mathbb{I} \end{bmatrix}$$
$$\boldsymbol{W}_K = \begin{bmatrix} \mathbf{0} & \mathbf{0} & C\mathbb{I} & \mathbf{0} \end{bmatrix}$$
$$\boldsymbol{W}_V = \begin{bmatrix} \mathbf{0} & \mathbf{0} & \mathbf{0} & \mathbf{0} \\ \mathbf{0} & 1 & \mathbf{0} & \mathbf{0} \\ \mathbf{0} & \mathbf{0} & \mathbf{0} & \mathbf{0} \\ \mathbf{0} & \mathbf{0} & \mathbf{0} & \mathbf{0} \end{bmatrix}$$

The pair $\boldsymbol{W}_Q$ and $\boldsymbol{W}_K$ attend tokens to the token directly to the right. The value matrix simply filters only the second row in-place. A second head can used to clear the original 1s, resulting in

$$X = \begin{bmatrix} x_1^{(1)} & \dots & x_1^{(j)} & x_2^{(j)} & \dots & x_{n-2}^{(j)} & = & y^{(j)} & x_1^{(j+1)} & \dots \\ 0 & \dots & 0 & 0 & \dots & 0 & 0 & 1 & 0 & \dots \\ \boldsymbol{p}_{n+1} & \dots & \boldsymbol{p}_{jn+1} & \boldsymbol{p}_{jn+2} & \dots & \boldsymbol{p}_{jn+n-2} & \boldsymbol{p}_{jn+n-1} & \boldsymbol{p}_{jn+n} & \boldsymbol{p}_{(j+1)n+1} & \dots \\ \boldsymbol{p}_n & \dots & \boldsymbol{p}_{jn} & \boldsymbol{p}_{jn+1} & \dots & \boldsymbol{p}_{jn+n-3} & \boldsymbol{p}_{jn+n-2} & \boldsymbol{p}_{jn+n-1} & \boldsymbol{p}_{(j+1)n} & \dots \end{bmatrix}, \tag{5}$$

as desired. $\qquad\square$

**Implementation of functions.** To illustrate a set of operations that could be implemented with a transformer, we consider approximating functions as sums of ReLUs; we use a result from Bai et al. (2023b), which we present below.

**Definition 1** (Definition 12 in Bai et al. (2023b))**.** *A function $g : \mathbb{R}^k \to \mathbb{R}$ is $(\epsilon, R, M, C)$-approximable by sum of ReLUs, if there exists an "(M,C)-sum of ReLUs" function*

$$f_{M,C}(\boldsymbol{x}) = \sum_{m=1}^{M} c_m ReLU(\boldsymbol{a}_m^\top [\boldsymbol{x}; 1]) \text{ with } \sum_{m=1}^{M} |c_m| \leq C, \ \max_{m \in [M]} \|\boldsymbol{a}_m\|_1 \leq 1, \ \boldsymbol{a}_m \in \mathbb{R}^{k+1}, \ c_m \in \mathbb{R}$$

*such that $\sup_{\boldsymbol{x} \in [-R,R]^k} |g(\boldsymbol{x}) - f_{(M,C)}(\boldsymbol{x})| \leq \epsilon$.*

**Definition 2** (Definition A.1 in Bai et al. (2023b))**.** *We say a function $g : \mathbb{R}^k \to \mathbb{R}$ is $(R, C_l)$-smooth if for $s = \lceil (k-1)/2 \rceil + 2$, $g \in C^{25}$ on $[-R, R]^k$ and*

$$\sup_{\boldsymbol{x} \in [-R,R]^k} \left\| \nabla^i g(\boldsymbol{x}) \right\|_\infty = \sup_{\boldsymbol{x} \in [-R,R]^k} \max_{j1,\dots,j_i \in [k]} \left| \partial_{x_{j1},\dots,x_{ji}} g(\boldsymbol{x}) \right| \leq L_i$$

*for all $i = 0, 1, 2$ with $\max_{0 \leq i \leq s} L_i R^i \leq C_l$.*

---
[5] $C^i$ denotes that a function is $i$ times differentiable with continuous $i$-th derivative.

**Proposition 1** (Proposition A.1 in Bai et al. (2023b)). *For any $\epsilon > 0$, $R \geq 1$, $C_l > 0$, we have that: Any $(R, C_l)$-smooth function, $g : \mathbb{R} \to \mathbb{R}$ is $(\epsilon, R, M, C)$-approximable by sum of ReLUs (Definition 1) with $M \leq C(k)C_l^2 \log(1 + C_l\epsilon)/\epsilon^2$.*

**Lemma 2.** *For any function $g : \mathbb{R}^k \to \mathbb{R}$ that is $(R, C_l)$-smooth, there exists a transformer with two layers, one head and width $\mathcal{O}(\log(n) + d)$, where $d$ satisfies the requirements of Prop. 1, such that given as input*

$$
X = \begin{bmatrix} x_1^{(1)} & \dots & y^{(j-1)} & x_1^{(j)} & \dots & x_{n-2}^{(j)} & = & y^{(j)} & x_1^{(j+1)} & \dots \\ 0 & \dots & 1 & 0 & \dots & 0 & 0 & 1 & 0 & \dots \\ \mathbf{0} & \dots & \mathbf{0} & \mathbf{0} & \dots & \mathbf{0} & \mathbf{0} & \mathbf{0} & \mathbf{0} & \dots \end{bmatrix},
$$

*it outputs*

$$
X = \begin{bmatrix} x_1^{(1)} & \dots & y^{(j-1)} & x_1^{(j)} & \dots & x_{n-2}^{(j)} & = & y^{(j)} & x_1^{(j+1)} & \dots \\ * & \dots & * & * & \dots & * & * & * & * & \dots \\ \mathbf{0} & \dots & \tilde{g}(x_i^{(j-1)}) - y^{(j-1)} & * & \dots & * & * & \tilde{g}(x_i^{(j)}) - y^{(j)} & * & \dots \end{bmatrix}
$$

*where $|\tilde{g}(x) - g(x)| \leq \epsilon$ and for some $i \in [1, \dots, n-2]$.*

*Proof.* We consider that the positional encodings are added in the input and we have

$$
X = \begin{bmatrix} x_1^{(1)} & \dots & x_1^{(j)} & x_2^{(j)} & \dots & x_{n-2}^{(j)} & = & y^{(j)} & x_1^{(j+1)} & \dots \\ 0 & \dots & 0 & 0 & \dots & 0 & 0 & 1 & 0 & \dots \\ \mathbf{0} & \dots & \mathbf{0} & \mathbf{0} & \dots & \mathbf{0} & \mathbf{0} & \mathbf{0} & \mathbf{0} & \dots \\ 1 & \dots & 1 & 1 & \dots & 1 & 1 & 1 & 1 & \dots \\ p_{n+1} & \dots & p_{jn+1} & p_{jn+2} & \dots & p_{jn+n-2} & p_{jn+n-1} & p_{jn+n} & p_{(j+1)n+1} & \dots \\ p_{n+1-s} & \dots & p_{jn+1-s} & p_{jn+2-s} & \dots & p_{jn+n-2-s} & p_{jn+n-1-s} & p_{jn+n-s} & p_{(j+1)n+1-s} & \dots \end{bmatrix} \tag{6}
$$

where we fix some positional encodings $p_k$ where $p_k^\top p_k$ is larger than $p_k^\top p_l$ by some threshold for $k \neq l$. The encodings used here are the binary representations of $k \in \{-1, 1\}^{\log(mn)}$. Further, we consider 1s in the positions with the results of the task to differentiate the context of the task and the result of the task. Define $s = n - i$, the distance between the result and the associated value in the context.

In the first layer, we use the MLP's to create $\tilde{g}$ according to Proposition 1

$$
X = \begin{bmatrix} x_1^{(1)} & \dots & x_1^{(j)} & x_2^{(j)} & \dots & x_{n-2}^{(j)} & = & y^{(j)} & x_1^{(j+1)} & \dots \\ 0 & \dots & 0 & 0 & \dots & 0 & 0 & 1 & 0 & \dots \\ \tilde{g}(x_1^{(1)}) & \dots & \tilde{g}(x_1^{(j)}) & \tilde{g}(x_2^{(j)}) & \dots & \tilde{g}(x_{n-2}^{(j)}) & \tilde{g}(=) & \tilde{g}(y^{(j)}) & \tilde{g}(x_1^{(j+1)}) & \dots \\ \mathbf{0} & \dots & \mathbf{0} & \mathbf{0} & \dots & \mathbf{0} & \mathbf{0} & \mathbf{0} & \mathbf{0} & \dots \\ 1 & \dots & 1 & 1 & \dots & 1 & 1 & 1 & 1 & \dots \\ p_{n+1} & \dots & p_{jn+1} & p_{jn+2} & \dots & p_{jn+n-2} & p_{jn+n-1} & p_{jn+n} & p_{(j+1)n+1} & \dots \\ p_{n+1-s} & \dots & p_{jn+1-s} & p_{jn+2-s} & \dots & p_{jn+n-2-s} & p_{jn+n-1-s} & p_{jn+n-s} & p_{(j+1)n+1-s} & \dots \end{bmatrix} \tag{7}
$$

The next operation is a shift of the sequence of $\tilde{g}(\cdot)$'s to the right by $s$. This will align the desired output $\tilde{g}(x_i^{(j)})$ with the observed output $y^{(j)}$. Consider the following weight matrices

$$
W_Q = [\dots \quad 0 \quad \mathbf{0} \quad \mathbb{I}] \tag{8}
$$

$$
W_K = [\dots \quad 0 \quad C\mathbb{I} \quad \mathbf{0}] \tag{9}
$$

$$
W_V = \begin{bmatrix} \mathbf{0} & \mathbf{0} & \mathbf{0} & \dots & \mathbf{0} \\ \mathbf{0} & \mathbf{0} & \mathbf{0} & \dots & \mathbf{0} \\ \mathbf{0} & \mathbf{0} & \mathbb{I} & \dots & \mathbf{0} \\ \vdots & \vdots & \vdots & & \vdots \\ \mathbf{0} & \mathbf{0} & \mathbf{0} & \dots & \mathbf{0} \end{bmatrix} \tag{10}
$$

for some large constant $C$ to decrease error from the softmax attending to the incorrect tokens. This produces (within a small error induced by using a softmax)

$$(\boldsymbol{X}^\top \boldsymbol{W}_K^\top \boldsymbol{W}_Q \boldsymbol{X})_{i,j} = \boldsymbol{p}_{n+i}^\top \boldsymbol{p}_{n-s+j} \tag{11}$$

$$\sigma_S(\boldsymbol{X}^\top \boldsymbol{W}_K^\top \boldsymbol{W}_Q X)_{i,j} = \mathbb{1}_{\{n+i=n-s+j\}} = \mathbb{1}_{\{i=j-s\}} \tag{12}$$

$$\boldsymbol{W}_V \boldsymbol{X} = \begin{bmatrix} \cdots & \mathbf{0} & \mathbf{0} & \cdots & \mathbf{0} & \mathbf{0} & \mathbf{0} & \cdots \\ \cdots & 0 & 0 & \cdots & 0 & 0 & 0 & \cdots \\ \cdots & \tilde{g}(\boldsymbol{x}_1^{(j)}) & \tilde{g}(\boldsymbol{x}_2^{(j)}) & \cdots & \tilde{g}(\boldsymbol{x}_{n-2}^{(j)}) & \tilde{g}(\boldsymbol{=}) & \tilde{g}(\boldsymbol{y}^{(j)}) & \cdots \\ & \vdots & \vdots & & \vdots & \vdots & \vdots & \\ \cdots & \mathbf{0} & \mathbf{0} & \cdots & \mathbf{0} & \mathbf{0} & \mathbf{0} & \cdots \end{bmatrix} \tag{13}$$

$$\boldsymbol{W}_V \boldsymbol{X} \sigma_S(\boldsymbol{X}^\top \boldsymbol{W}_K^\top \boldsymbol{W}_Q \boldsymbol{X}) = \begin{bmatrix} \cdots & \mathbf{0} & \mathbf{0} & \cdots & \mathbf{0} & \mathbf{0} & \mathbf{0} & \cdots \\ \cdots & 0 & 0 & \cdots & 0 & 0 & 0 & \cdots \\ \cdots & * & * & \cdots & * & * & \tilde{g}(\boldsymbol{x}_i^{(j)}) & \cdots \\ & \vdots & \vdots & & \vdots & \vdots & \vdots & \\ \cdots & \mathbf{0} & \mathbf{0} & \cdots & \mathbf{0} & \mathbf{0} & \mathbf{0} & \cdots \end{bmatrix} \tag{14}$$

$$\boldsymbol{X} + \boldsymbol{W}_V \boldsymbol{X} \sigma_S(\boldsymbol{X}^\top \boldsymbol{W}_K^\top \boldsymbol{W}_Q \boldsymbol{X}) = \begin{bmatrix} \cdots & \boldsymbol{x}_1^{(j)} & \boldsymbol{x}_2^{(j)} & \cdots & \boldsymbol{x}_{n-1}^{(j)} & \boldsymbol{=} & \boldsymbol{y}^{(j)} & \cdots \\ \cdots & 0 & 0 & \cdots & 0 & 0 & 1 & \cdots \\ \cdots & * & * & \cdots & * & * & \tilde{g}(\boldsymbol{x}_i^{(j)}) & \cdots \\ \cdots & \mathbf{0} & \mathbf{0} & \cdots & \mathbf{0} & \mathbf{0} & \mathbf{0} & \cdots \\ \cdots & 1 & 1 & \cdots & 1 & 1 & 1 & \cdots \\ \cdots & \boldsymbol{p}_{jn+1} & \boldsymbol{p}_{jn+2} & \cdots & \boldsymbol{p}_{jn+n-2} & \boldsymbol{p}_{jn+n-1} & \boldsymbol{p}_{jn+n} & \cdots \\ \cdots & \boldsymbol{p}_{jn+1-s} & \boldsymbol{p}_{jn+2-s} & \cdots & \boldsymbol{p}_{jn+n-2-s} & \boldsymbol{p}_{jn+n-1-s} & \boldsymbol{p}_{jn+n-s} & \cdots \end{bmatrix} \tag{15}$$

$$\tag{16}$$

Each matrix above only shows the slice that contains the $j$-th in-context example. This is repeated for each of the other in-context examples.

As a final step with an MLP, subtract row 1 from row 3 to achieve the following output:

$$\begin{bmatrix} \cdots & \boldsymbol{x}_1^{(j)} & \boldsymbol{x}_2^{(j)} & \cdots & \boldsymbol{x}_{n-1}^{(j)} & \boldsymbol{=} & \boldsymbol{y}^{(j)} & \cdots \\ \cdots & 0 & 0 & \cdots & 0 & 0 & 1 & \cdots \\ \cdots & * & * & \cdots & * & * & \tilde{g}(\boldsymbol{x}_i^{(j)}) - \boldsymbol{y}^{(j)} & \cdots \\ \cdots & \mathbf{0} & \mathbf{0} & \cdots & \mathbf{0} & \mathbf{0} & \mathbf{0} & \cdots \\ \cdots & 1 & 1 & \cdots & 1 & 1 & 1 & \cdots \\ \cdots & \boldsymbol{p}_{jn+1} & \boldsymbol{p}_{jn+2} & \cdots & \boldsymbol{p}_{jn+n-2} & \boldsymbol{p}_{jn+n-1} & \boldsymbol{p}_{jn+n} & \cdots \\ \cdots & \boldsymbol{p}_{jn+1-s} & \boldsymbol{p}_{jn+2-s} & \cdots & \boldsymbol{p}_{jn+n-2-s} & \boldsymbol{p}_{jn+n-1-s} & \boldsymbol{p}_{jn+n-s} & \cdots \end{bmatrix} \tag{17}$$

$$\square$$

**Copy Tasks**  As has been experimentally investigated, the situation where a specific position within the context is copied as the label can be easily implemented by setting $g(\boldsymbol{x}) = \boldsymbol{x}$. The dependence on the subscript $i$ within the construction is what allows the position copied to vary.

### D.2.1 IDENTIFYING IF TASK'S OUTPUT MATCHES THE IN-CONTEXT EXAMPLE

**Lemma 3.** *A three layer transformer with ReLU MLPs and embedding dimension $\mathcal{O}(d + \log(mn))$ can calculate the proportion of in context examples that come from a specific task, where $m$ is the number of in-context examples, each of length $n$ and dimension $d$.*

*Proof.* We now have a matrix of the following form.

$$
\begin{bmatrix}
\dots & \boldsymbol{x}_1^{(j)} & \boldsymbol{x}_2^{(j)} & \dots & = & \boldsymbol{y}^{(j)} & \dots \\
\dots & 0 & 0 & \dots & 0 & 1 & \dots \\
\dots & * & * & \dots & * & f(\boldsymbol{x}^{(j)}) - \boldsymbol{y}^{(j)} & \dots \\
\dots & \boldsymbol{0} & \boldsymbol{0} & \dots & \boldsymbol{0} & \boldsymbol{0} & \dots \\
\dots & 1 & 1 & \dots & 1 & 1 & \dots \\
\dots & \boldsymbol{p}_{jn+1} & \boldsymbol{p}_{jn+2} & \dots & \boldsymbol{p}_{jn+n-1} & \boldsymbol{p}_{jn+n} & \dots \\
\dots & \boldsymbol{p}_{jn+1-s} & \boldsymbol{p}_{jn+2-s} & \dots & \boldsymbol{p}_{jn+n-1-s} & \boldsymbol{p}_{jn+n-s} & \dots
\end{bmatrix}
\tag{18}
$$

If the task is correct, than $f(\boldsymbol{x}^{(j)}) - \boldsymbol{y}^{(j)} \approx \boldsymbol{0}$, with some small error coming from softmaxs and function approximation error. First, we find the $L1$-norm of $f(\boldsymbol{x}^{(j)}) - \boldsymbol{y}^{(j)}$ using an MLP. For calculating $\|\boldsymbol{z}\|_1$ for arbitrary $\boldsymbol{z}$, we can use

$$
\|\boldsymbol{z}\|_1 = \sum_{i=1}^{d} \text{ReLU}(\boldsymbol{z}_i) - \text{ReLU}(-\boldsymbol{z}_i)
\tag{19}
$$

which can be done in a single 1-layer MLP. Thus, we have

$$
\begin{bmatrix}
\dots & \boldsymbol{x}_1^{(j)} & \boldsymbol{x}_2^{(j)} & \dots & = & \boldsymbol{y}^{(j)} & \dots \\
\dots & 0 & 0 & \dots & 0 & 1 & \dots \\
\dots & * & * & \dots & * & f(\boldsymbol{x}^{(j)}) - \boldsymbol{y}^{(j)} & \dots \\
\dots & * & * & \dots & * & \|f(\boldsymbol{x}^{(j)}) - \boldsymbol{y}^{(j)}\|_1 & \dots \\
\dots & \boldsymbol{0} & \boldsymbol{0} & \dots & \boldsymbol{0} & \boldsymbol{0} & \dots \\
\dots & 1 & 1 & \dots & 1 & 1 & \dots \\
\dots & \boldsymbol{p}_{jn+1} & \boldsymbol{p}_{jn+2} & \dots & \boldsymbol{p}_{jn+n-1} & \boldsymbol{p}_{jn+n} & \dots \\
\dots & \boldsymbol{p}_{jn+1-s} & \boldsymbol{p}_{jn+2-s} & \dots & \boldsymbol{p}_{jn+n-1-s} & \boldsymbol{p}_{jn+n-s} & \dots
\end{bmatrix}
\tag{20}
$$

Notice that if some task has different dimension than another task, the "extra" rows would be zero and will not affect the result.

For clarity, we set all $*$ values in the $\|\cdot\|_1$ row to 1s. These will cause the following $\hat{\delta}$ in the following set these to 0. This operation can be omitted as the construction handles these trash values at a later layer.

Let $b$ represent the value of the flag in the second row marking the $\boldsymbol{y}$ vectors and let $x$ represent the values in the row with $\|f(\boldsymbol{x}^{(j)}) - \boldsymbol{y}^{(j)}\|_1$. The following ReLUs set the * values to 1.

$$
x \leftarrow x + 1 - \text{ReLU}(x - Cb) - \text{ReLU}(Cb - C + 1)
\tag{21}
$$

for some large constant $C$. When $b = 0$, this reduces to $x + 1 - x = 1$, and when $b = 1$, this reduces to $x + 1 - 1 = x$, as desired.

$$
\begin{bmatrix}
\dots & \boldsymbol{x}_1^{(j)} & \boldsymbol{x}_2^{(j)} & \dots & = & \boldsymbol{y}^{(j)} & \dots \\
\dots & 0 & 0 & \dots & 0 & 1 & \dots \\
\dots & * & * & \dots & * & f(\boldsymbol{x}^{(j)}) - \boldsymbol{y}^{(j)} & \dots \\
\dots & 1 & 1 & \dots & 1 & \|f(\boldsymbol{x}^{(j)}) - \boldsymbol{y}^{(j)}\|_1 & \dots \\
\dots & \boldsymbol{0} & \boldsymbol{0} & \dots & \boldsymbol{0} & \boldsymbol{0} & \dots \\
\dots & 1 & 1 & \dots & 1 & 1 & \dots \\
\dots & \boldsymbol{p}_{jn+1} & \boldsymbol{p}_{jn+2} & \dots & \boldsymbol{p}_{jn+n-1} & \boldsymbol{p}_{jn+n} & \dots \\
\dots & \boldsymbol{p}_{jn+1-s} & \boldsymbol{p}_{jn+2-s} & \dots & \boldsymbol{p}_{jn+n-1-s} & \boldsymbol{p}_{jn+n-s} & \dots
\end{bmatrix}
\tag{22}
$$

Now define a thresholding function $\hat{\delta}(z)$ that satisfies $\hat{\delta}(0) = 1$ and $\hat{\delta}(z) = 0$ for $z >> 0$. One such function used here is

$$
\hat{\delta}_C(z) = \text{ReLU}(1 - Cz)
\tag{23}
$$

for some constant $C$, where larger $C$ captures a narrower neighborhood of 0.

However, a slight change needs to be added to $\hat{\delta}_C$. In the same row as $\|f(\boldsymbol{x}^{(j)}) - \boldsymbol{y}^{(j)}\|$ are many values that need to be discarded. Let $b$ be the bit for the current column marking if the column contains an $\boldsymbol{x}$ or a $\boldsymbol{y}$. We use instead

$$\hat{\delta}_C(b, z) = \text{ReLU}(b - Cz) \tag{24}$$

This will be zero whenever $b = 0$ and $z \geq 0$. We then have as output

$$
\begin{bmatrix}
\dots & \boldsymbol{x}_1^{(j)} & \boldsymbol{x}_2^{(j)} & \dots & = & \boldsymbol{y}^{(j)} & \dots \\
\dots & 0 & 0 & \dots & 0 & 1 & \dots \\
\dots & * & * & \dots & * & f(\boldsymbol{x}^{(j)}) - \boldsymbol{y}^{(j)} & \dots \\
\dots & 1 & 1 & \dots & 1 & \|f(\boldsymbol{x}^{(j)}) - \boldsymbol{y}^{(j)}\|_1 & \dots \\
\dots & 0 & 0 & \dots & 0 & \hat{\delta}_C(\|f(\boldsymbol{x}^{(j)}) - \boldsymbol{y}^{(j)}\|_1) & \dots \\
\dots & \mathbf{0} & \mathbf{0} & \dots & \mathbf{0} & \mathbf{0} & \dots \\
\dots & 1 & 1 & \dots & 1 & 1 & \dots \\
\dots & \boldsymbol{p}_{jn+1} & \boldsymbol{p}_{jn+2} & \dots & \boldsymbol{p}_{jn+n-1} & \boldsymbol{p}_{jn+n} & \dots \\
\dots & \boldsymbol{p}_{jn+1-s} & \boldsymbol{p}_{jn+2-s} & \dots & \boldsymbol{p}_{jn+n-1-s} & \boldsymbol{p}_{jn+n-s} & \dots
\end{bmatrix} \tag{25}
$$

Importantly, $\hat{\delta}_C(\|f(\boldsymbol{x}^{(j)}) - \boldsymbol{y}^{(j)}\|_1) = 1$ when $f(\cdot)$ is the correct task and $\hat{\delta}_C(\|f(\boldsymbol{x}^{(j)}) - \boldsymbol{y}^{(j)}\|_1) = 0$ when $f(\cdot)$ disagrees by more than $\frac{1}{C}$ in $L1$-norm.

Lastly, for the next step in the construction, we need to average these soft indicators $\hat{\delta}$ to see how common $f$ is within the context. This is done with an attention layer. Let $\boldsymbol{W}_Q$ select the row with all 1s multiplied by some large constant C, and let $\boldsymbol{W}_K$ select the row with flags for results $\boldsymbol{y}$. Then

$$
\boldsymbol{X}^\top \boldsymbol{W}_K^\top \boldsymbol{W}_Q \boldsymbol{X} =
\begin{bmatrix}
\vdots & \vdots & & \vdots & \vdots \\
0 & 0 & \dots & 0 & 0 \\
0 & 0 & \dots & 0 & 0 \\
\vdots & \vdots & & \vdots & \vdots \\
0 & 0 & \dots & 0 & 0 \\
C & C & \dots & C & C \\
\vdots & \vdots & & \vdots & \vdots
\end{bmatrix} \tag{26}
$$

$$
\sigma_S(\boldsymbol{X}^\top \boldsymbol{W}_K^\top \boldsymbol{W}_Q \boldsymbol{X}) \approx
\begin{bmatrix}
\vdots & \vdots & & \vdots & \vdots \\
0 & 0 & \dots & 0 & 0 \\
0 & 0 & \dots & 0 & 0 \\
\vdots & \vdots & & \vdots & \vdots \\
0 & 0 & \dots & 0 & 0 \\
1/m & 1/m & \dots & 1/m & 1/m \\
\vdots & \vdots & & \vdots & \vdots
\end{bmatrix} \tag{27}
$$

where a $1/m$ will appear in every row corresponding to a result $\boldsymbol{y}$. Let the value matrix select the row containing $\hat{\delta}(\|f(\boldsymbol{x}^{(j)} - \boldsymbol{y}^{(j)})\|_1)$. Denote $p = \frac{1}{m} \sum_{j=1}^m \hat{\delta}(\|f(\boldsymbol{x}^{(j)} - \boldsymbol{y}^{(j)})\|_1)$. Without causal masking, we would have as output

$$
\begin{bmatrix}
\dots & \boldsymbol{x}_1^{(j)} & \boldsymbol{x}_2^{(j)} & \dots & \boldsymbol{=} & \boldsymbol{y}^{(j)} & \dots \\
\dots & 0 & 0 & \dots & 0 & 1 & \dots \\
\dots & * & * & \dots & * & f(\boldsymbol{x}^{(j)}) - \boldsymbol{y}^{(j)} & \dots \\
\dots & 1 & 1 & \dots & 1 & \|f(\boldsymbol{x}^{(j)}) - \boldsymbol{y}^{(j)}\|_1 & \dots \\
\dots & 0 & 0 & \dots & 0 & \hat{\delta}_C(\|f(\boldsymbol{x}^{(j)}) - \boldsymbol{y}^{(j)}\|_1) & \dots \\
\dots & p & p & \dots & p & p & \dots \\
\dots & \mathbf{0} & \mathbf{0} & \dots & \mathbf{0} & \mathbf{0} & \dots \\
\dots & 1 & 1 & \dots & 1 & 1 & \dots \\
\dots & \boldsymbol{p}_{jn+1} & \boldsymbol{p}_{jn+2} & \dots & \boldsymbol{p}_{jn+n-1} & \boldsymbol{p}_{jn+n} & \dots \\
\dots & \boldsymbol{p}_{jn+1-s} & \boldsymbol{p}_{jn+2-s} & \dots & \boldsymbol{p}_{jn+n-1-s} & \boldsymbol{p}_{jn+n-s} & \dots
\end{bmatrix}
\tag{28}
$$

However, with causal masking, we can only guarantee that $p$ will appear in the columns containing the most recent example being queried. Thankfully, this is all that is needed.  □

### D.3 TASK EXECUTION

**Lemma 4.** *A two layer transformer, with embedding dimension $\mathcal{O}(d + \log(mn))$ can perform a task and weight its output by the proportion of examples of that task seen within the context.*

Now that the proportions of each task have been identified in the context, the task itself needs to be executed for the new example being queried. To simplify notation, let the input to this step be

$$
\boldsymbol{X} =
\begin{bmatrix}
\dots & \boldsymbol{x}_1^{(m)} & \boldsymbol{x}_2^{(m)} & \dots & \boldsymbol{x}_{n-2}^{(m)} & \boldsymbol{=} \\
\dots & * & * & \dots & * & * \\
\dots & p & p & \dots & p & p \\
\dots & \mathbf{0} & \mathbf{0} & \dots & \mathbf{0} & \mathbf{0} \\
\dots & * & * & \dots & * & * \\
\dots & 1 & 1 & \dots & 1 & 1
\end{bmatrix}
\tag{29}
$$

Following the same process as outlined above, although with slightly different positional encodings, calculate $f(\boldsymbol{x}^{(m)})$ and place that result in the final column being decoded. These need to be added at the beginning of the construction, but are only introduced here for clarity.

$$
\begin{bmatrix}
\dots & \boldsymbol{x}_1^{(m)} & \boldsymbol{x}_2^{(m)} & \dots & \boldsymbol{x}_{n-2}^{(m)} & \boldsymbol{x}_{n-1}^{(m)} \\
\dots & * & * & \dots & * & * \\
\dots & p & p & \dots & p & p \\
\dots & * & * & \dots & * & f(\boldsymbol{x}^{(m)}) \\
\dots & \mathbf{0} & \mathbf{0} & \dots & \mathbf{0} & \mathbf{0} \\
\dots & * & * & \dots & * & * \\
\dots & 1 & 1 & \dots & 1 & 1 \\
\dots & 0 & 0 & \dots & 0 & 1 \\
\dots & 0 & 0 & \dots & 1 & 0
\end{bmatrix}
\tag{30}
$$

We will transform the row containing all $p$ to be able to approximately multiply $p$ by $f(\boldsymbol{x}^{(m)})$. Using the second to last row, perform $p \to 1 - p$. Using the last two rows, clear out the rest of that row and fill it with $-C$ for some large constant $C$. We then have

$$
\begin{bmatrix}
\dots & \boldsymbol{x}_1^{(m)} & \boldsymbol{x}_2^{(m)} & \dots & \boldsymbol{x}_{n-2}^{(m)} & \boldsymbol{x}_{n-1}^{(m)} \\
\dots & * & * & \dots & * & * \\
\dots & -C & -C & \dots & p & 1-p \\
\dots & * & * & \dots & * & f(\boldsymbol{x}^{(m)}) \\
\dots & \mathbf{0} & \mathbf{0} & \dots & \mathbf{0} & \mathbf{0} \\
\dots & * & * & \dots & * & * \\
\dots & 1 & 1 & \dots & 1 & 1 \\
\dots & 0 & 0 & \dots & 0 & 1 \\
\dots & 0 & 0 & \dots & 1 & 0
\end{bmatrix}
\tag{31}
$$

Further, use the second-to-last row to clear out all $*$ in the rows below.

$$
\begin{bmatrix}
\dots & \boldsymbol{x}_1^{(m)} & \boldsymbol{x}_2^{(m)} & \dots & \boldsymbol{x}_{n-2}^{(m)} & \boldsymbol{x}_{n-1}^{(m)} \\
\dots & * & * & \dots & * & * \\
\dots & -C & -C & \dots & p & 1-p \\
\dots & \mathbf{0} & \mathbf{0} & \dots & \mathbf{0} & f(\boldsymbol{x}^{(m)}) \\
\dots & \mathbf{0} & \mathbf{0} & \dots & \mathbf{0} & \mathbf{0} \\
\dots & * & * & \dots & * & * \\
\dots & 1 & 1 & \dots & 1 & 1 \\
\dots & 0 & 0 & \dots & 0 & 1 \\
\dots & 0 & 0 & \dots & 1 & 0
\end{bmatrix}
\tag{32}
$$

These previous operations can all be done in a single MLP.

Lastly, use an attention layer where $\boldsymbol{W}_K$ selects the row with the $-C$s, $\boldsymbol{W}_Q$ selects the row with all 1s, and $\boldsymbol{W}_V$ selects the $f(\boldsymbol{x}^{(m)})$. For the last token $\boldsymbol{x}_L$,

$$
\boldsymbol{X}^\top \boldsymbol{W}_K^\top \boldsymbol{W}_Q \boldsymbol{x}_L =
\begin{bmatrix}
\vdots \\ -C \\ -C \\ \vdots \\ p \\ 1-p
\end{bmatrix}
[1] =
\begin{bmatrix}
\vdots \\ -C \\ -C \\ \vdots \\ p \\ 1-p
\end{bmatrix}
\tag{33}
$$

$$
\sigma_S(\boldsymbol{X}^\top \boldsymbol{W}_K^\top \boldsymbol{W}_Q \boldsymbol{x}_L) \approx
\begin{bmatrix}
\vdots \\ -\infty \\ -\infty \\ \vdots \\ p \\ 1-p
\end{bmatrix}
=
\begin{bmatrix}
\vdots \\ 0 \\ 0 \\ \vdots \\ \frac{1}{1+e^{1-2p}} \\ 1 - \frac{1}{1+e^{1-2p}}
\end{bmatrix}
\tag{34}
$$

$$
\boldsymbol{W}_V \boldsymbol{x}_L \sigma_S(\boldsymbol{X}^\top \boldsymbol{W}_K^\top \boldsymbol{W}_Q \boldsymbol{x}_L) =
\begin{bmatrix}
\vdots \\ \mathbf{0} \\ \frac{1}{1+e^{1-2p}}\mathbf{0} + (1 - \frac{1}{1+e^{1-2p}})f(\boldsymbol{x}^{(m)}) \\ \mathbf{0} \\ \vdots
\end{bmatrix}
\tag{35}
$$

$$
\boldsymbol{x}_L + \boldsymbol{W}_V \boldsymbol{x}_L \sigma_S(\boldsymbol{X}^\top \boldsymbol{W}_K^\top \boldsymbol{W}_Q \boldsymbol{x}_L) =
\begin{bmatrix}
\boldsymbol{x}_{n-1}^{(m)} \\ * \\ 1-p \\ \frac{1}{1+e^{1-2p}}f(\boldsymbol{x}^{(m)}) \\ \mathbf{0} \\ * \\ 1 \\ 1 \\ 0
\end{bmatrix}
\tag{36}
$$

Importantly, we are left with $\frac{1}{1+e^{1-2p}}f(\boldsymbol{x}^{(m)})$. The factor $\frac{1}{1+e^{1-2p}}$ is approximately $p$, especially around $\frac{1}{2}$. This multiplication can also be calculated more accurately with approximations using ReLUs or sigmoids, but for brevity and following experimental evidence of a sigmoid shape in task superpositions, these options are ommited.

## D.4 SUPERPOSED TASKS WITH PARALLEL HEADS

The above construction works for a single task, where the output is weighted by the proportions of the task within the context. To complete the construction of a transformer that does superposition of tasks, each of these models needs to be placed within the same overall transformer. This is described here.

Let there be a collection of tasks $\{t_i\}_{i=1}^{T}$ which can be executed by transformers with model weights represented by subscripts $(\cdot_i)$. With the input to each transformer being $\boldsymbol{X}^{(i)}$, the overall input matrix is given by vertically stacking these matrices.

$$
\boldsymbol{X} = \begin{bmatrix} \boldsymbol{X}_1 \\ \boldsymbol{X}_2 \\ \vdots \\ \boldsymbol{X}_{T-1} \\ \boldsymbol{X}_T \end{bmatrix}
\tag{37}
$$

Similarly, define each MLP's weights and biases as

$$
\boldsymbol{W} = \mathrm{diag}(\boldsymbol{W}_1, \ldots, \boldsymbol{W}_T) \quad \boldsymbol{b} = \begin{bmatrix} \boldsymbol{b}_1 \\ \vdots \\ \boldsymbol{b}_T \end{bmatrix}
\tag{38}
$$

This puts every MLP to be independent of each other. Lastly, we need to change the attention layers. This requires the use of one head per task. In each of the following, $\boldsymbol{W}^{(i)}$ is a weight matrix for head $i$, $(\boldsymbol{W})_i$ is the weight matrix for task $i$ in its individual transformer, and each matrix below is in the $i$-th block.

$$
\boldsymbol{W}_V^{(i)} = \begin{bmatrix} \vdots \\ \boldsymbol{0} \\ (\boldsymbol{W}_V)_i \\ \boldsymbol{0} \\ \vdots \end{bmatrix}^{\top} \qquad \boldsymbol{W}_K^{(i)} = \begin{bmatrix} \vdots \\ \boldsymbol{0} \\ (\boldsymbol{W}_K)_i \\ \boldsymbol{0} \\ \vdots \end{bmatrix}^{\top} \qquad \boldsymbol{W}_Q^{(i)} = \begin{bmatrix} \vdots \\ \boldsymbol{0} \\ (\boldsymbol{W}_Q)_i \\ \boldsymbol{0} \\ \vdots \end{bmatrix}^{\top}
\tag{39}
$$

In all, this model executes multiple tasks in superposition by using parallel streams of heads that each performs a single task. Task identification can happen through the same mechanism as task execution by comparing the output of the task on each in context example with the true output.

For context related tasks, there needs to be positional encodings that allow for looking back a fixed number of tokens. For context agnostic tasks, a wide MLP can be used to approximate arbitrary non-linear transformations of the input. Each of these tasks only require a small number of layers, significantly smaller than those of modern LLMs. It may be possible that LLMs do certain tasks with different combinations of layers.

Also, if we take the feature $p$ from each parallel stream, this creates the following task identifier.

$$
\boldsymbol{v} = \begin{bmatrix} p_1 \\ p_2 \\ \vdots \\ p_T \end{bmatrix}
\tag{40}
$$

Interpolating between the pure tasks, represented by unit vectors, different amounts of each task will appear in the superposition in roughly equal proportions to those found in $\boldsymbol{v}$.

Lastly, we restate this construction formally.

**Theorem 1.** *A seven layer transformer with embedding dimension $\mathcal{O}(d + \log(mn))$ with $K$ heads per attention layer can perform $k$ tasks on vectors of dimension $d$ in superposition, with weighting based on $m$ different in-context examples each of length $n$ .*

*Proof.* Using in succession each of Lemma 1, Lemma 2, Lemma 3, and Lemma 4, a transformer with the desired properties can execute $k$ tasks in parallel. Lemma 1 identifies positions within the context that contain the labels $\boldsymbol{y}$. Lemma 2 then uses function approximation to perform arbitrary tasks within the architecture, which are then used by 3 to find the proportions of each task and aggregate them into a single task identifier. Lastly, Lemma 4 uses this task identifier to create a weighted sum of outputs from the different tasks based on their in-context proportions. $\square$

**Remark.** *Transformers of greater depth than seven layers can also represent this construction by setting the weights in all other layers for the non residual part to zero.*

# E    ADDITIONAL EXPERIMENT AND ANALYSIS OF TASK SUPERPOSITION IN PRETRAINED MODELS

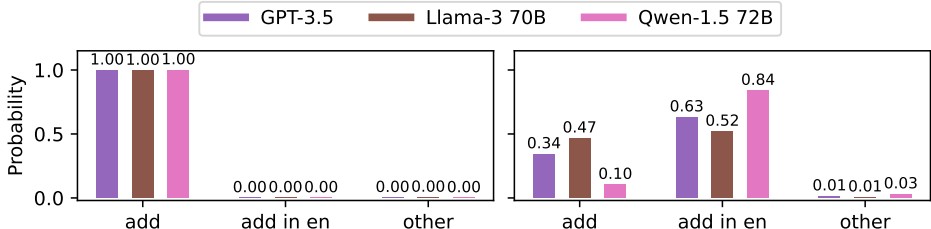

(a) **(left)** All task examples from task `add`; **(right)** equal number of task examples from `add` and `add_in_en`.

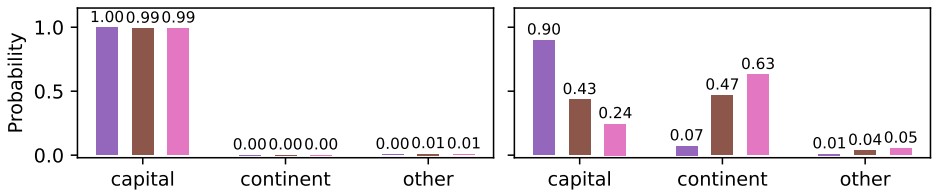

(b) **(left)** All task examples from task `capital`; **(right)** equal number of task examples from `capital` and `continent`.

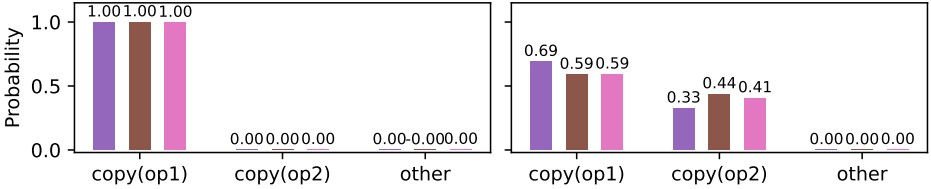

(c) **(left)** All task examples from task `copy(op1)`; **(right)** equal number of task examples from `copy(op1)` and `copy(op2)`.

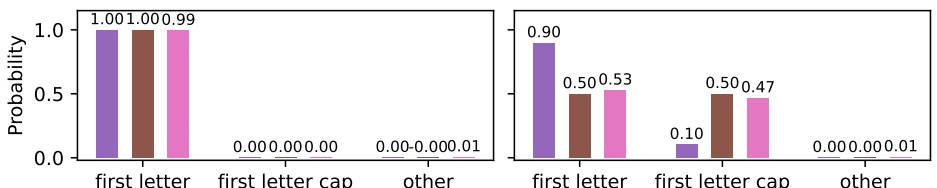

(d) **(left)** All task examples from task `first_letter`; **(right)** equal number of task examples from `first_letter` and `first_letter_cap`.

Figure 7: For each subplot, we consider two tasks `task1` and `task2` and on the left side we plot the median probability for task answers where all task examples in prompts are from `task1`; on the right side we plot the median probability for task answer where task examples from `task1` and from `task2` in prompts are equal.

## E.1    IS SUPERPOSITION REALLY HAPPENING?

In Figure 2 we plot distributions of probabilities for each task in four settings. Note that in setting 3 (corresponds to Figure 2c), the `other` category has low probabilities for all models but for settings 1, 2 and 4 (correspond to Figure 2a, 2b and 2d respectively), the `other` category is not always low. A natural question to ask is:

> *For the non-zero probabilities we observe on each task answer, are they indications of task superposition or by-products of prediction noise?*

We set up an experiment to investigate this. For each setting, we select two tasks `task1` and `task2`; then we consider two scenarios: (1) we provide prompts where all task examples come from `task1` and (2) we provide prompts where half of the task examples come from `task1` and the other half of task examples come from `task2`; in both scenarios we measure the probabilities for task answers of `task1` and `task2` and see how these probabilities change between scenario (1) and (2). For each scenario, we test it on 100 prompts (each task has 10 in-context examples) and plot the median of task answers in Figure 7. As is shown in Figure 7 left side, where there is no task example from `task2` in the prompt, the probabilities for task answers of `task2` are near 0 for all models; on the right side, where there is an equal number of task examples that are from `task1` and `task2`, the probabilities for task answers of `task2` increase significantly. This indicates that when we provide prompts that mix task examples from different tasks, the prediction on each task answer is more than just pure prediction noise.

### E.2 MORE ANALYSIS ON `OTHER` CATEGORY IN MODEL'S OUTPUT DISTRIBUTION

In Figure 2, in settings 1, 2 and 4 (correspond to Figure 2a, 2b and 2d respectively), since there are also non-negligible probabilities on the `other` category (and it is a summation of all other probabilities), we further investigate the probabilities in `other` category. In particular, for each prompt, we use beam search (where we stop searching when we encounter "\n") to find answers that have top-$(K + 1)$ probabilities and record the maximum probability of the answer that is not one of the task answers. In Figure 8, we plot the median of such probability along with medians of probabilities of each task answer.

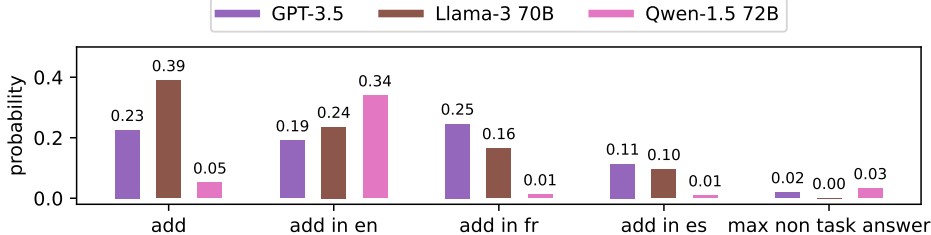

(a) Setting 1: Addition in original numerical form and in different languages.

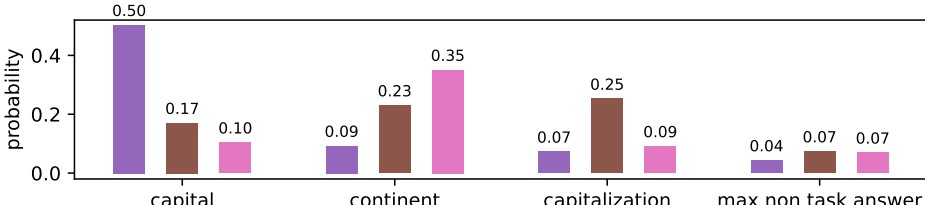

(b) Setting 2: Capital name, continent name and capitalization.

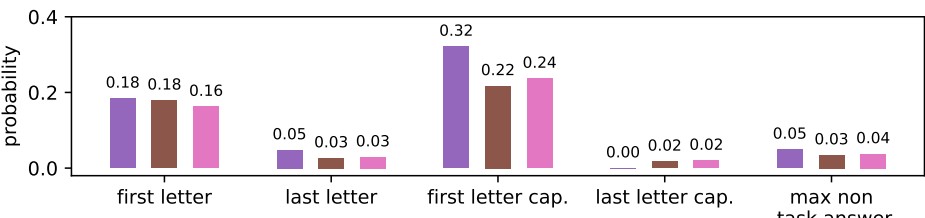

(c) Setting 4: First or last letter in upper or lower cases.

Figure 8: For each subplot, we plot the medians of probabilities for each task answer and the median of the maximum probability of the answer that is not one of the task answers.

In Figure 8a (setting 1), for GPT-3.5 and Llama-3, we can observe that the medians of the probabilities for most probable non-task-answer are significantly lower than that of each task answer. This indicates that the models are effectively performing task superposition across the four tasks, with any other

answer having a low probability. For Qwen-1.5, however, the median probabilities of task answers for `add`, `add_in_fr`, `add_in_es` and the most probable non-task-answer are relatively low. This may be attributed to the model's limited ability to perform the `add_in_fr` and `add_in_es` tasks.

In Figure 8b (setting 2), we see that for all models, the median probability of the most probable non-task-answer is lower than that of each task answer. A possible explanation for the most probable non-task-answer still having a probability around 0.07 is the presence of related tasks: (1) `CAPITAL`, which returns the capital of a country in uppercase, (2) `CONTINENT`, which returns the continent of a country in uppercase, and (3) `identity`, which directly returns the country name. If these task answers are excluded when calculating the top-$(K + 1)$ probabilities, and the maximum probability of the non-task-answer is recalculated, the median probability drops to less than 0.02 for Qwen-1.5 and Llama-3, and less than 0.001 for GPT-3.5. This suggests that while the models may not always strictly perform task superposition on the provided tasks, they may perform compositions of the tasks. We think it is an interesting direction for the future work to study the relation between task superposition and task composition.

In Figure 8c (setting 4), we notice that the medians of the maximum non-task-answer probability are slightly higher than (or comparable to) those of the task answers for `last_letter` and `last_letter_cap`. This could be due to the fact that these tasks are more challenging for the models. On the other hand, the medians of most probable non-task-answer probability are significantly lower than that of task answers for `first_letter` and `first_letter_cap`, where we consider the model is performing task superposition on.

### E.3 MEASURING ACCURACY IN TASK SUPERPOSITION

We further investigate how task superposition affect task performance. In particular, for a prompt consisting examples of $K$ tasks (each task has an equal number of task examples), we define a task being correctly performed if its task answer lies in the top-$K$ answers (that we use beam search to find). We compare the accuracy against individual task accuracy where we provide prompts consisting of task examples of only 1 task and define the task being correctly performed if the task answer is the top-1 answer.

We calculate accuracy in $K = 1$ case and $K > 1$ case using 100 prompts and show the result in Table 2. We find that as we increase the number of tasks, all models exhibit an accuracy decrease in correctly performing each individual task. Notably, however, the accuracy degradation of Llama3-70B is less than that of Llama2-70B across most tasks. We believe this is indicative that improved model training techniques lead to better preservation of task superposition,

| Model | $K = 1$ | | | | $K = 4$ | | | |
|---|---|---|---|---|---|---|---|---|
| | t1 | t2 | t3 | t4 | t1 | t2 | t3 | t4 |
| GPT-3.5 | 100 | 99 | 85 | 91 | 95 $(-5)$ | 90 $(-9)$ | 84 $(-1)$ | 84 $(-7)$ |
| Llama-3 70B | 100 | 99 | 97 | 99 | 100 $(0)$ | 99 $(0)$ | 96 $(-1)$ | 92 $(-7)$ |
| Llama-2 70B | 100 | 96 | 69 | 77 | 88 $(-12)$ | 96 $(0)$ | 63 $(-6)$ | 46 $(-31)$ |
| Qwen-1.5 72B | 100 | 94 | 52 | 70 | 66 $(-34)$ | 91 $(-3)$ | 28 $(-24)$ | 34 $(-36)$ |

(a) Setting 1: Addition in original numerical form and in different languages where t1 = add, t2 = add_in_en, t3 = add_in_fr, t4 = add_in_es.

| Model | $K = 1$ | | | $K = 3$ | | |
|---|---|---|---|---|---|---|
| | t1 | t2 | t3 | t1 | t2 | t3 |
| GPT-3.5 | 100 | 100 | 100 | 93 $(-7)$ | 62 $(-38)$ | 56 $(-44)$ |
| Llama-3 70B | 100 | 100 | 100 | 82 $(-18)$ | 90 $(-10)$ | 83 $(-17)$ |
| Llama-2 70B | 97 | 94 | 90 | 75 $(-22)$ | 75 $(-19)$ | 50 $(-40)$ |
| Qwen-1.5 72B | 100 | 99 | 91 | 65 $(-35)$ | 87 $(-12)$ | 48 $(-43)$ |

(b) Setting 2: Naming the capital, continent and capitalize the country name where t1 = capital, t2 = continent, t3 = capitalization.

| Model | $K = 1$ | | | $K = 3$ | | |
|---|---|---|---|---|---|---|
| | t1 | t2 | t3 | t1 | t2 | t3 |
| GPT-3.5 | 100 | 100 | 100 | 100 $(0)$ | 97 $(-3)$ | 97 $(-3)$ |
| Llama-3 70B | 100 | 100 | 100 | 99 $(-1)$ | 100 $(0)$ | 99 $(-1)$ |
| Llama-2 70B | 100 | 100 | 95 | 95 $(-5)$ | 99 $(-1)$ | 84 $(-11)$ |
| Qwen-1.5 72B | 100 | 100 | 99 | 100 $(0)$ | 98 $(-2)$ | 98 $(-1)$ |

(c) Setting 3: t1 = copy(op1), t2 = copy(op2) and t3 = op1+op2.

| Model | $K = 1$ | | | | $K = 4$ | | | |
|---|---|---|---|---|---|---|---|---|
| | t1 | t2 | t3 | t4 | t1 | t2 | t3 | t4 |
| GPT-3.5 | 100 | 87 | 100 | 54 | 94 $(-6)$ | 56 $(-31)$ | 97 $(-3)$ | 12 $(-42)$ |
| Llama-3 70B | 100 | 63 | 100 | 40 | 99 $(-1)$ | 29 $(-34)$ | 99 $(-1)$ | 13 $(-27)$ |
| Llama-2 70B | 100 | 55 | 100 | 38 | 99 $(-1)$ | 35 $(-20)$ | 97 $(-3)$ | 7 $(-31)$ |
| Qwen-1.5 72B | 100 | 62 | 100 | 34 | 89 $(-11)$ | 30 $(-32)$ | 100 $(0)$ | 15 $(-19)$ |

(d) Setting 4: First or last letter in upper or lower cases where t1 = first_letter, t2 = last_letter, t3 = first_letter_cap, t4 = last_letter_cap.

Table 2: Accuracy for each task in percentage, with the delta change given in parenthesis. For each setting we calculate the accuracy with prompts consisting of task examples of only one task ($K = 1$ case) and with prompts consisting of examples from multiple tasks ($K > 1$ case).

## F    ADDITIONAL FIGURES

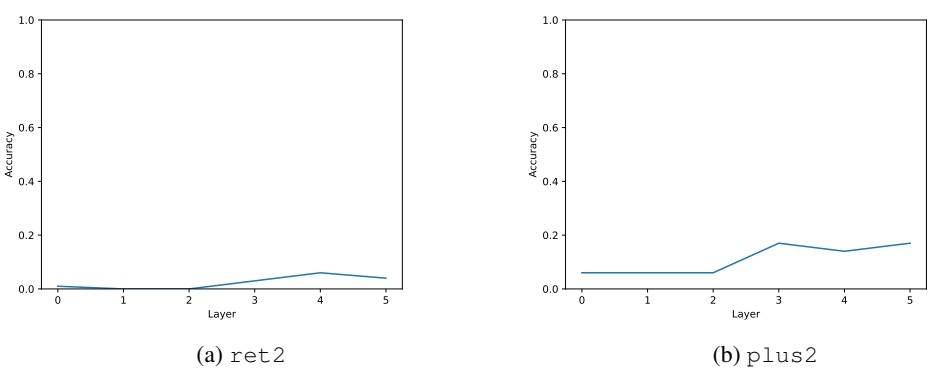

(a) ret2

(b) plus2

Figure I: Accuracy for each choice of the intermediate layer $\ell$ on task ret2 and plus2.

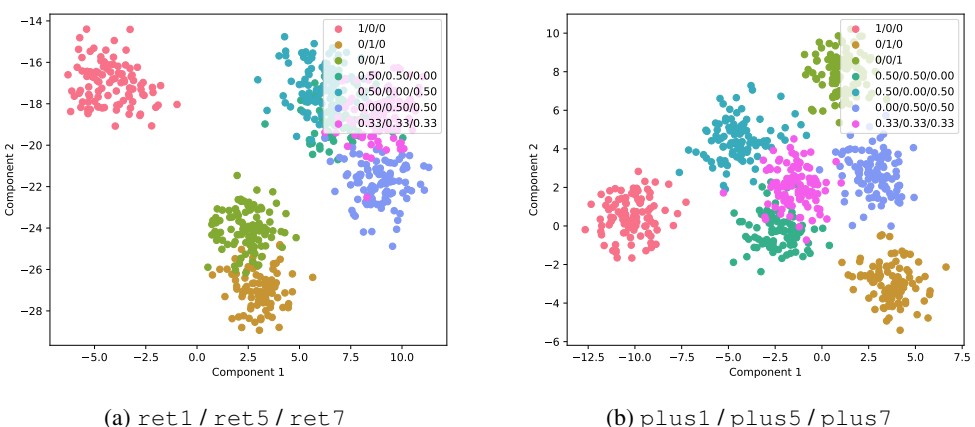

(a) ret1 / ret5 / ret7

(b) plus1 / plus5 / plus7

Figure II: Task vectors projected onto two axes chosen by LDA for two sets of tasks: **(a)** ret1, ret5 and ret7 and **(b)** plus1, plus5 and plus7.

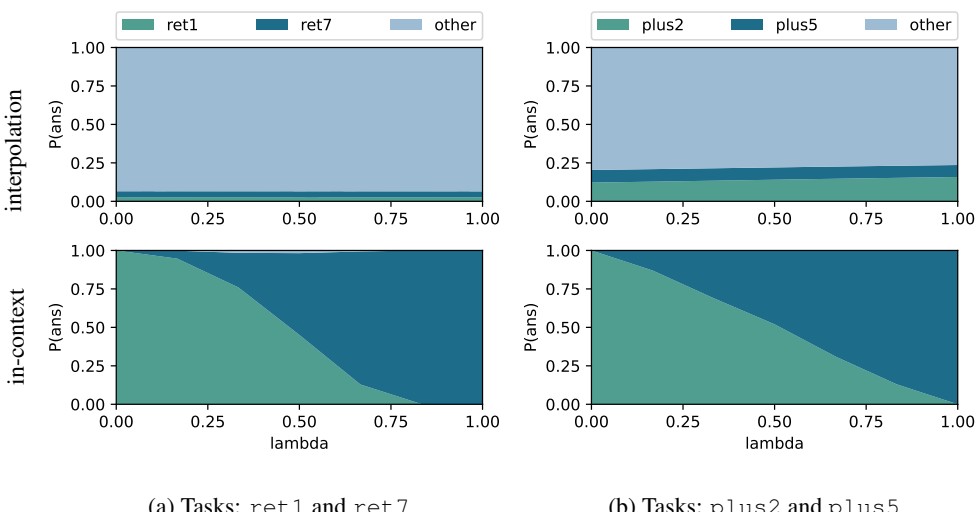

(a) Tasks: `ret1` and `ret7`

(b) Tasks: `plus2` and `plus5`

Figure III: We vary the proportion, $\lambda$, between two tasks and observe how the output probabilities for the correct answers change. The proportion $\lambda$ is varied in two ways: (1) in the top row, we plot the output from patching in a convex combination of task vectors for two tasks. (2) in the bottom row, we plot the output from a mixed proportion of in-context examples for the two tasks. Subplot (a) shows the output probabilities from mixing two retrieval tasks and (b) shows the probabilities from mixing two addition tasks.

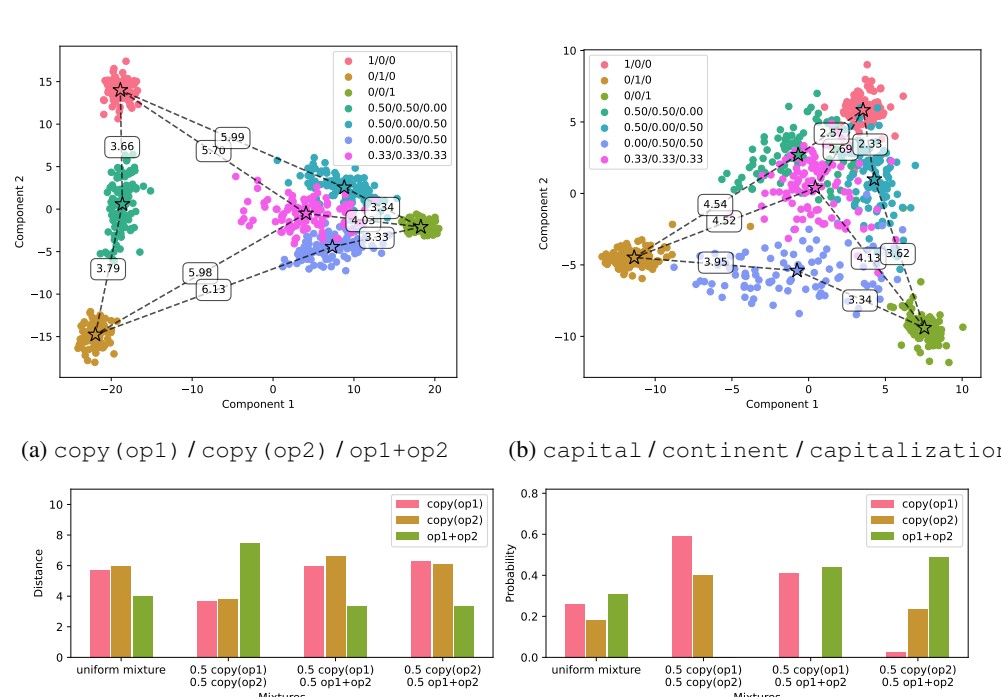

(a) `copy(op1)` / `copy(op2)` / `op1+op2`  (b) `capital` / `continent` / `capitalization`

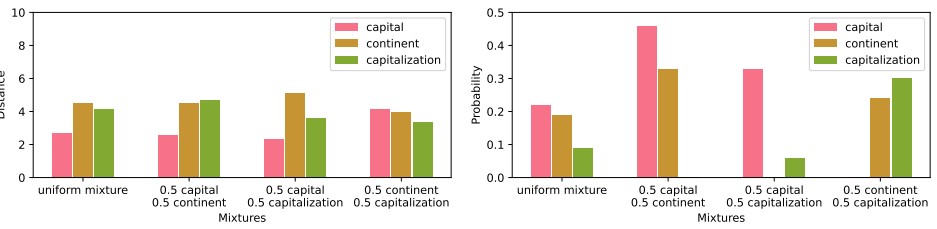

(c) **(left)** Euclidean distances between centroids and **(right)** corresponding medians of the task answer probability in setting of (a).

(d) **(left)** Euclidean distances between centroids and **(right)** corresponding medians of the task answer probability in setting of (b).

Figure IV: Task vectors of Llama-3 8B projected onto two axes chosen by LDA for two sets of tasks: **(a)** `copy(op1)`, `copy(op2)` and `op1+op2` and **(b)** `capital`, `continent` and `capitalization`; centroids and Euclidean distance between centroids of mixtures and centroids of non-mixtures are labeled. In **(c)** and **(d)**, left side shows Euclidean distances from centroids of each mixture clusters to centroids of non-mixture clusters, and right side shows medians of probability for each task answer when provided with prompts of task mixture.

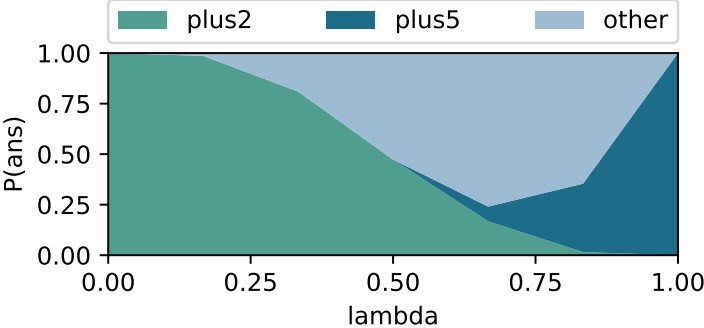

Figure V: We vary the proportion, $\lambda$, between two tasks and observe how the output probabilities of a 1-head transformer for the correct answers change. The proportion $\lambda$ is varied on two tasks: `plus2` and `plus5`.

