# OpenReview forum: "Everything Everywhere All at Once: LLMs can In-Context Learn Multiple Tasks in Superposition"
_ICLR.cc/2025/Conference — Submitted to ICLR 2025_

### Official Review · Reviewer_Q86s · 2024-10-27

**Soundness:** 3
**Presentation:** 3
**Contribution:** 2
**Rating:** 6
**Confidence:** 3

**Summary:**

This paper studies the "task superposition" phenomenon in ICL, i.e.,
LLMs can perform multiple different ICL tasks simultaneously.
Empirically, it is observed that the phenomenon is ubiquitous across different LLMs and larger models can handle more tasks simultaneously
and better represent the distribution of in-context tasks.
Theoretically, the authors show Transformers are expressive enough for task superposition.
They also relate task superposition to internal combinations of task vectors.

**Strengths:**

1. The paper introduces the interesting "task superposition" phenomenon, which provides insights on the ICL mechanism of LLMs.

2. The paper reveals an impressive correspondence between task superposition and internal combinations of task vectors.

3. The paper theoretically proves Transformers are expressive enough for task superposition.

**Weaknesses:**

1. The paper only illustrates the superposition phenomenon in the synthetic tasks. It seems unclear whether real-world tasks have the superposition phenomenon
or how the superposition phenomenon affects LLMs in practice.

2. The theoretical explanation only considers the expressive power of Transformers. Since it has been proved that Transformers are Turing-complete,
the theoretical result does not provide sufficient insights.

**Questions:**

1. Theorem 1 states a quantitative relation between the embedding dimension, the numbers of heads, tasks, in-context examples, and example length.
Does this quantitative relation reflect how the model hyperparameters influence the task superposition capability in practice? For example,
to perform $K$ tasks, does the model need at least $K$ heads?

2. Could it be possible to prove the learnability of a Transformer that has the task superposition capability? Some works [1] [2] prove the learnability of ICL
under the Bayesian model of ICL.

3. Are there any insights on how the superposition phenomenon of the LLM influences real-world tasks? Is it desired or undesired?
In the examples of the paper, an LLM with the superposition phenomenon seems to choose a random task if there are multiple different ICL tasks simultaneously.
Can this be disadvantageous in some scenarios? Could it be possible to control which task the LLM will perform?

[1] Zhang, Yufeng, Fengzhuo Zhang, Zhuoran Yang, and Zhaoran Wang. "What and how does in-context learning learn? bayesian model averaging, parameterization, and generalization." arXiv preprint arXiv:2305.19420 (2023).

[2] Wies, Noam, Yoav Levine, and Amnon Shashua. "The learnability of in-context learning." Advances in Neural Information Processing Systems 36 (2024).

---

> ### Author Response · Authors · 2024-11-22
> **Response to Reviewer Q86s (1/2)**
>
> We thank the reviewer for their thoughtful feedback on our paper. Below, we address the specific questions raised by the reviewer.
>
> **W1&Q3: Task superposition in practice**
> > In the examples of the paper, an LLM with the superposition phenomenon seems to choose a random task if there are multiple different ICL tasks simultaneously.
>
> We would first like to clarify that task superposition is not just "choosing a random task if there are multiple different ICL tasks". Rather, task superposition is the phenomenon that when provided with in-context task examples from multiple tasks, the model will output non-negligible probabilities on the corresponding task answers (we provide a more detailed explanation of task superposition framework in the "**A more explicit definition of task and task superposition**" section of our [general response](https://openreview.net/forum?id=FxLxbJTm7F&noteId=AZ0veisbFn)). Below we further discuss the contribution of our work.
>
> > It seems unclear whether real-world tasks have the superposition phenomenon or how the superposition phenomenon affects LLMs in practice.
> > Are there any insights on how the superposition phenomenon of the LLM influences real-world tasks? Is it desired or undesired? Can this be disadvantageous in some scenarios? Could it be possible to control which task the LLM will perform?
>
> We appreciate reviewer's thoughtful questions regarding the practicality and implications of task superposition. We acknowledge that the immediate practical applications of task superposition may not be fully apparent. However, we believe that our findings contribute significantly to the fundamental understanding of LLMs and open avenues for future research in several important areas.
>
> 1. Task superposition could have practical benefits in scenarios where an LLM serves multiple users simultaneously. By leveraging superposition (e.g., using a better decoding strategy that can maintain the model’s multi-task state throughout the generation process), an LLM might efficiently handle diverse tasks in parallel, improving scalability and responsiveness in LLMs serving [3] under multi-user environment.
> 2. The phenomenon of task superposition has intriguing implications for AI alignment. It suggests that alignment interventions might influence the distribution over latent skills conditioned on the prompt (i.e.,  $\mathbb{P}(\texttt{skill} | \texttt{prompt})$ ) without necessarily affecting the model's performance conditioned on a specific skill and input (i.e.,  $\mathbb{P}(\texttt{answer} | \texttt{skill}, \texttt{prompt})$ ). This raises questions about the effectiveness of alignment strategies that focus solely on modifying model outputs without considering the underlying superposition of skills. While this is speculative, we believe it highlights a critical area for future research in understanding and improving alignment methodologies.
> 3. Our observations align with the "simulator-in-superposition" hypothesis [1,2] that emerged with the advent of GPT-3. This hypothesis suggests that LLMs can simulate multiple potential continuations or behaviors simultaneously, reflecting a superposition of different skills or tasks. By demonstrating that LLMs can internally represent and process multiple tasks in parallel when provided with mixed in-context examples, we provide empirical support for this theoretical framework.
> 4. We believe that explicitly characterizing the phenomenon of task superposition is valuable in its own right. Adding to our understanding of how LLMs work contributes to the broader field of AI research.
>
> Our goal with this paper is to shed light on an emergent behavior of LLMs that has not been previously documented, and we think is surprising. We believe that understanding such phenomena is crucial for the research of LLMs.

---

> ### Author Response · Authors · 2024-11-22
> **Response to Reviewer Q86s (2/2)**
>
> **W2: Transformers are Turing-complete and the theoretical result does not provide sufficient insights**
>
> We *respectfully disagree* with the reviewer on this. Previous works [4,5] have indeed established that Transformers, with access to infinite memory, are Turing complete. However, while Turing completeness implies broad theoretical capabilities, it does not address the depth or width required to achieve the specific result presented in our paper. In fact, based on the results of [4,5], the required depth and/or width would likely be impractically large, and adapting existing results to our specific setting may not be straightforward.
>
> Given this context, we emphasize that the primary contribution of our work lies in the empirical observations that drive our study. The theoretical results are included to complement these observations, offering a plausible explanatory framework, but they are not the primary focus of our novelty.
>
>
> **Q1: Does the quantitative relation in Thm 1 reflect how the model hyperparameters influence the task superposition capability in practice? For example, to perform K tasks, does the model need at least K heads?**
>
> In our Section 7, we conduct experiments to show that larger models have better task superposition capability, that they can solve more tasks in parallel and better calibrate to ICL distribution (Finding 5).
>
> We conduct additional experiments to investigate whether we need $K$ heads for $K$ tasks superposition. In particular, we trained a 1-head small model on addition tasks (the setting in Section 4) and tested on prompts with mixture of in-context examples of `plus2` and `plus5`. In [Figure V](https://github.com/76N5/superposition/blob/main/Figure_V.pdf), we observe that while the 1-head model can in-context learn a single task (that corresponds to two end sides where $\lambda=0$ and $1$), it cannot perform task superposition when provided with examples of two tasks. For example, when $\lambda=0.5$ when we provide prompt with equal number of task examples from `plus2` and `plus5`, half of the output probability mass lies on the `plus2` task answer but the output probability for `plus5` task answer is almost 0. This indicates that we need at least $2$ heads for $2$ tasks superposition.
>
>
> **Q2: Learnability of a Transformer that has the task superposition capability**
>
> Thanks for your suggestion. We think showing the learnability of a Transformer that has the task superposition capability is indeed a valuable and interesting direction for future work.
>
> **References**
>
> [1] Reynolds, L., & McDonell, K. (2021). Multiversal views on language models. arXiv preprint arXiv:2102.06391.
>
> [2] moire. Language models are multiverse generators, January 2021. https://generative.ink/posts/language-models-are-multiverse-generators
>
> [3] Kwon, W., Li, Z., Zhuang, S., Sheng, Y., Zheng, L., Yu, C. H., ... & Stoica, I. (2023, October). Efficient memory management for large language model serving with pagedattention
>
> [4]: Jorge Pérez, Pablo Barceló, and Javier Marinkovic. 2021. Attention is turing complete. J. Mach. Learn. Res. 22, 1, Article 75 (January 2021), 35 pages.
>
> [5]: Giannou, Angeliki, et al. "Looped transformers as programmable computers." International Conference on Machine Learning. PMLR, 2023.

---

### Official Review · Reviewer_nCnF · 2024-11-03

**Soundness:** 3
**Presentation:** 2
**Contribution:** 2
**Rating:** 6
**Confidence:** 4

**Summary:**

The paper studies the mechanism of in-context learning with multiple tasks. It discovers the “task superposition” of LLMs. Using both theory and pretraining experiments on toy tasks, it demonstrates the ability of LLMs to implement the “task superposition”. They also explore the composition of different task vectors and how the model size affects the “task superposition” ability.

**Strengths:**

1. The paper designs tasks to study the ICL with multiple tasks in LLMs.
2. The composition of task vectors is interesting.

**Weaknesses:**

Overall, the paper provides multiple findings, but removing redundant findings and implementing more analysis of interesting findings would be better.

1. Some findings are redundant. Findings 1 and 5 are not surprising given previous works such as Bai et al. [2023].
2. Findings 2-4 are interesting. However, more work is required on these findings. A similar “task vectors” analysis could also be applied to finding 2. Findings 3 and 4 can also relate to the non-clibrated results in Figure 2. It would be better to emphasize these findings.

Yu Bai, Fan Chen, Huan Wang, Caiming Xiong, and Song Mei. Transformers as statisticians: Provable in-context learning with in-context algorithm selection, 2023b.

**Questions:**

1. Can authors demonstrate their novelty? Bai et al. [2023] have shown that “A single transformer can adaptively select different base ICL algorithms—or even perform qualitatively different tasks—on different input sequences, without any explicit prompting of the right algorithm or task.” This paper considers the mixture of different tasks with varying parameters of mixture. It is equivalent to selecting the correct algorithm that fits the mixture probability in the in-context examples.
2. Figure 2 presents that the output of LLMs is not calibrated with the in-context examples. How does this finding interplay with the results in Section 6, where the composition of task vectors is not proportional to the composition of in-context examples?

---

> ### Author Response · Authors · 2024-11-22
> **Response to Reviewer nCnF (1/2)**
>
> We thank the reviewer for their thoughtful feedback on our paper. Below, we address the specific questions raised by the reviewer.
>
> **W1&Q1: Difference between our work and the work by Bai et al.**
>
> > This paper considers the mixture of different tasks with varying parameters of mixture. It is equivalent to selecting the correct algorithm that fits the mixture probability in the in-context examples.
>
> We acknowledge the valuable insights from Bai et al. [1] and plan to expand our discussion of these connections in the revised paper. However, we *respectfully disagree* with the characterization that our work is equivalent to algorithm selection in Bai et al. While there are connections, the works differ in several aspects.
>
> **1. Settings**:
> * Bai et al. focuses on algorithm selection where during test time Bai et al. tests transformers using in-context examples from one task, and shows that the model selects the right algorithm to solve one target task.
> * We test the model with multiple qualitatively different ICL tasks presented in a single prompt, and it solves multiple tasks in superposition.
>
> **2. Training Requirements:**
> * In Bai et al.'s empirical setting, for the algorithm that the model selects during test time, it requires explicit training on that algorithm.
> * While it is true that superposition can be think of as an algorithm that outputs probability distribution that calibrates with in-context examples, applying Bai et al.'s setting here assume we have explicitly trained the model on all such algorithm that calibrates output distribution with respect to in-context examples, i.e., the model was trained on data where the $\texttt{prompt}$ mixes examples from qualitatively different tasks and labels being non one-hot to calibrate the in-context task examples.
>     * This is likely not the case in the pre-trained model since the labels during LLMs training are in one-hot format (probability 1 on one token and 0 elsewhere); furthermore, prompts consisting of task examples from different tasks (within a single prompt) are unlikely to be part of some training data.
>     * In fact, in Finding 2 we find that we don't need to train the model on such data, and the model can do task superposition even if we train the model to in-context learn one task at a time.
>
> **3. Scaling Experiment:**
> * To our best knowledge, our Finding 5 that studies model's task superposition capability as the model scales was also not observed in other works.
>
> We would be happy to further clarify these distinctions if the reviewer finds the current explanation insufficient.
>
> [1] Yu Bai, Fan Chen, Huan Wang, Caiming Xiong, and Song Mei. Transformers as statisticians: Provable in-context learning with in-context algorithm selection, 2023b.

---

> ### Author Response · Authors · 2024-11-22
> **Response to Reviewer nCnF (2/2)**
>
> **W2\&Q2: More analysis on Findings**
> > A similar “task vectors” analysis could also be applied to finding 2.
>
> We extract task vectors from our small trained-from-scratch models using the same pipeline as the pretrained models. While our task vector extraction works well for large, pretrained models, we could not find task vectors that work well for our small models. For example, in [Figure I](https://github.com/76N5/superposition/blob/main/Figure_I.pdf) we plot the accuracy on task `ret2` and `plus2` when using vectors extracted from different layers and observe that the maximum accuracy we can get is lower than 0.2 (while for large pretrained model such accuracy is usually near 1).
>
> We plot the task vectors decomposition in [Figure II](https://github.com/76N5/superposition/blob/main/Figure_II.pdf) and conduct the linear interpolation experiment in [Figure III](https://github.com/76N5/superposition/blob/main/Figure_III.pdf). We notice that task vectors of uniform mixtures for retrieval tasks do not lie in the middle of three individual task vector clusters and convex combinations in both cases do not produce task superposition. A possible explanation is that, while current task vectors are extracted from a specific layer, for the small models, the feature that represents a task is likely not localized to a specific layer (as indicated in [Figure I](https://github.com/76N5/superposition/blob/main/Figure_I.pdf)), which means that we need to modify how we extract the task vectors for small models. We believe this is an important area for further research.
>
>
> > Figure 2 presents that the output of LLMs is not calibrated with the in-context examples. How does this finding interplay with the results in Section 6, where the composition of task vectors is not proportional to the composition of in-context examples?
>
> There is a correlation between the inter-cluster distances of task vectors in section 6 and calibration of LLM output probabilities in Figure 2. To illustrate this, we plotted the task vector scatter plot in the style of Figure 4 alongside the distances between centroids of the clusters and the corresponding probabilities on task answers in [Figure IV](https://github.com/76N5/superposition/blob/main/Figure_IV.pdf). We observe that in most cases, for task vectors of mixtures, a closer distance to task vectors of a task gives higher probability in the corresponding task answer. In particular,
> * The model's preference of `op1+op2` over `copy(op1)` or `copy(op2)` corresponds to smaller distance from mixture cluster centroid to the `op1+op2` cluster centroid.
> * The model's preference of `capital` over `continent` and `capitalization` corresponds to smaller distance from mixture cluster centroid to the `capital` cluster centroid.
>
> That said, we do not believe this relationship is necessarily proportional or linear. For example, in [Figure IV\(c)](https://github.com/76N5/superposition/blob/main/Figure_IV.pdf), for the `0.5 copy(op1), 0.5 op1+op2` mixture, it is significantly closer to `op1+op2` cluster than to `copy(op2)` cluster, but the model only assigns slightly more probability to `op1+op2` task answer than to `copy(op2)` task answer. So, while we show that the model composes task vectors internally, it is *more than a convex combination of task vectors*. It will be an interesting future research area to capture the mechanism of how model mixes the task vectors to produce superposed answers.
>
> (We will also incorporate this analysis in our future revised manuscript).

---

### Official Review · Reviewer_P2uz · 2024-11-04

**Soundness:** 3
**Presentation:** 3
**Contribution:** 2
**Rating:** 5
**Confidence:** 4

**Summary:**

The paper studied a specific type of LLMs in-context learning task, which the authors termed “task superposition”.  The study is comprehensive and extensive, including theoretical analysis and in-depth empirical investigation.

**Strengths:**

1. The paper is clearly written and well presented.
2. The investigation towards the proposed ability is comprehensive and in-depth.

**Weaknesses:**

Due to the inherit hierarchical nature of the concept “task”, I think the definition of “task” is at the core of this line of work. For example, one can argue Fig 1(a) essentially only contains a single task, which is addition regardless of language, or one can also argue that it contains only two tasks, which are numerical addition and multi-lingual translation, or four tasks as defined in the paper.

Hence, with a slight change of perspective (i.e., one abstract level up), the so-called superposition of tasks ICL itself basically defines a single new task. Since we know that LLMs can do ICL for a single task, it is not surprising that LLMs can also do the (single) task of “superposition of tasks” through ICL. Therefore, the term “tasks superposition” is misleading and unclear.

Some suggestions:
- Provide a clearer definition of what constitutes a "task" in the framework.
- More detailed discussion on how the notion of task superposition differs from or adds to existing understandings of in-context learning.
- Address how task superposition is distinct from a more complex single task.

**Questions:**

1. How do you generate the output probabilities for the answers (e.g., in Fig 1(a))? Do you generate multiple outputs for a fixed prompt, then filter out the wrong ones, and count the correct ones over multiple prompts?
2. What does it mean by "perform multiple tasks in a single inference call”, and “solve tasks in parallel”? What exactly is this “a single inference call”, is there an example?

---

> ### Author Response · Authors · 2024-11-22
> **Response to Reviewer P2uz (1/2)**
>
> We thank the reviewer for their thoughtful feedback on our paper. Below, we address the specific questions raised by the reviewer.
>
> **W1: Clearer definition of task/task superposition and how it differs from existing understandings of ICL**
>
> We add a more explicit formulation of task and task superposition and explains how it differs from existing understandings of ICL in the "**A more explicit definition of task and task superposition**" section of our [general response](https://openreview.net/forum?id=FxLxbJTm7F&noteId=AZ0veisbFn). We would happy to further explain if the reviewer finds this unclear.
>
>
> **W2: Distinction from a more complex single task**
>
> Under our new definition and framework in the [general response](https://openreview.net/forum?id=FxLxbJTm7F&noteId=AZ0veisbFn), a more complex single task can be think of as a vector of multiple simple tasks. For example, let $g_1, g_2:\mathcal{X}\rightarrow \mathcal{Y}$ be two simple tasks, we have a $g_3=(g_1, g_2):\mathcal{X}\rightarrow \mathcal{Y}\times\mathcal{Y}$ as a more complex task.
>
> Here is a concrete example. Let $\mathcal{X}, \mathcal{Y}$ be integers. Let $g_1$ be $g_1(x)=x+3$ (plus-three task) and $g_2(x)=x-1$ (minus-one task). In the existing in-context learning framework, we form prompts where task examples are from a single task:
>
> $\texttt{prompt}_1=$`1->4; 2->5; 3->6; 4->7; 5->` (task examples from $g_1$)
>
> $\texttt{prompt}_2=$`1->0; 2->1; 3->2; 4->3; 5->` (task examples from $g_2$)
>
> and if we provide two prompts to language model $\texttt{LM}$, we can get $\mathbb{P}(8\mid\texttt{prompt}_1)=0.98$ and $\mathbb{P}(4\mid\texttt{prompt}_2)=0.99$. We can observe that $\texttt{LM}$ assigns most of the probability mass on a single task answer in both cases and this indicates $\texttt{LM}$ in-context learns $g_1$ and $g_2$ respectively.
>
> In our setting, we form prompt where task examples are from different tasks
>
> $\texttt{prompt}\_{\text{mix}}=$`1->4; 2->1; 3->2; 4->7; 5->` (task examples from $g_1$ and $g_2$)
>
> and $\texttt{LM}$ gives $\mathbb{P}(8\mid\texttt{prompt}\_{\text{mix}})=0.49$ and $\mathbb{P}(4\mid\texttt{prompt}\_{\text{mix}})=0.48$, putting two significant portions of probability mass on two task answers respectively. We call this phenomenon task superposition, and we can observe this from a single forward pass because we can get both $\mathbb{P}(8\mid\texttt{prompt}\_{\text{mix}})$ and $\mathbb{P}(4\mid\texttt{prompt}\_{\text{mix}})$ from $\texttt{LM}(\texttt{prompt}\_{\text{mix}})$, which is a probability distribution over $\mathcal{V}$. This indicates that $\texttt{LM}$ performs two tasks **simultaneously** when provided with $\texttt{prompt}\_{\text{mix}}$.
>
> We can also consider a more complex single task $g_3=(g_1, g_2)$ and form the prompt
>
> $\texttt{prompt}_3=$`1->4,0; 2->5,1; 3->6,2; 4->7,3; 5->` (task examples from $g_3$)
>
> and note that task examples are all from a single task $g_3$, we will get $\mathbb{P}((8,4)\mid \texttt{prompt}_3)=0.96$ from $\texttt{LM}$. Since $\texttt{LM}$ predicts next token autoregressively, the full process would be
> 1. $\mathbb{P}($`8`$\mid$`1->4,0; 2->5,1; 3->6,2; 4->7,3; 5->`$)=0.99$. Append `8` to the prompt.
> 2. $\mathbb{P}($`,`$\mid$`1->4,0; 2->5,1; 3->6,2; 4->7,3; 5->8`$)=0.99$. Append `,` to the prompt.
> 3. $\mathbb{P}($`4`$\mid$`1->4,0; 2->5,1; 3->6,2; 4->7,3; 5->8,`$)=0.98$, and we get model's output `8,4` with probability $0.99\times0.99\times 0.98=0.96$ (or an additional step where the model predicts an empty space indicating the end symbol).
>
> Here the model is still performing a single complex task using multiple steps (each step performing a single simple sub-task) where the model puts most of its probability mass to a single token at each step. We highlight that this "more complex single task" setting is different from our task superposition setting as this setting first calculate $4$ and then calculate $8$ in different steps (in different model forward passes).
>
> We will incorporate this discussion in our revised manuscript.

---

> ### Author Response · Authors · 2024-11-22
> **Response to Reviewer P2uz (2/2)**
>
> **Q1: How do you generate model output probabilities?**
>
> We first clarify that, given $K$ tasks $g_1,...,g_K$ and a $\texttt{prompt}$ consisting of $m$ task examples (from $K$ tasks) and a query $x^{(m+1)}$, we calculate the output probabilities of task answers $g_1(x^{(m+1)}),...,g_K(x^{(m+1)})$ by calculating $\mathbb{P}(g_1(x^{(m+1)})\mid\texttt{prompt}),...,\mathbb{P}(g_K(x^{(m+1)})\mid\texttt{prompt})$.
>
> If task answers are single tokens, then this can be calculated in a forward pass as $\texttt{LM}$ outputs a distribution over $\mathcal{V}$ given $\texttt{prompt}$. If task answers are multi-tokens, we calculate each $\mathbb{P}(g_i(x^{(m+1)})\mid\texttt{prompt}), i=1,...,K$ autoregressively. That is, if in the token space $\texttt{prompt}=[v_1,...,v_M]$ has $M$ tokens, $g_i(x^{(m+1)})=[u_1,...,u_N]$ has $N$ tokens, $v_j, u_k\in\mathcal{V}, j\in[M], k\in[N]$, then
> \\begin{aligned}
> \mathbb{P}(g_i(x^{(m+1)})\mid\texttt{prompt})&=\mathbb{P}([u_1,...,u_N]\mid[v_1,...,v_M])\\\\
> &=\mathbb{P}(u_1\mid[v_1,...,v_M])\prod_{j=2}^N\mathbb{P}(u_j\mid[v_1,...,v_M, u_1,...,u_{j-1}]).
> \\end{aligned}
>
> Our Appendix B formalizes on this, please let us know if you would like us to extend it further.
>
> **Q2: More explanation on "perform multiple tasks in a single call"**
>
> We provide an example in our [response to W2](https://openreview.net/forum?id=FxLxbJTm7F&noteId=YKuvxCxjTq), where we show that given $\texttt{prompt}\_{\text{mix}}$ with task examples from two tasks, the model puts significant portion of probability mass on two task answers respectively. We would happy to further explain if the reviewer finds this unclear.
>
> We would also like to note that when task answers are multi-tokens, the model is still doing multiple tasks in parallel when provided with $\texttt{prompt}$ and predicting the first token by putting non-negligible probability mass on first tokens of each task answers. On the other hand, as we discussed in the "Limitation and Future Directions" section of paper, using existing decoding strategy, after the first token is generated, the model tends to converge on predicting tokens for a single task. We show that the model has the capability for multi-task execution and highlights a critical area for future research -- developing decoding strategies that can maintain the model’s multi-task state throughout the generation process (e.g., recent "superposed decoding" [1] offers some hope towards this direction).
>
> [1] Shen, E., Fan, A., Pratt, S. M., Park, J. S., Wallingford, M., Kakade, S. M., ... & Kusupati, A. (2024). Superposed Decoding: Multiple Generations from a Single Autoregressive Inference Pass. arXiv preprint arXiv:2405.18400.

---

### Official Review · Reviewer_Ssa4 · 2024-11-04

**Soundness:** 4
**Presentation:** 4
**Contribution:** 3
**Rating:** 6
**Confidence:** 4

**Summary:**

This paper demonstrates evidence of task “superposition” in LLMs when doing in-context learning, i.e. that LLMs can perform, or predict tokens for, multiple tasks simultaneously when shown those tasks in-context.

The first major argument for this is an empirical study of both pre-trained and from-scratch transformers. The authors first show that, across 3 pre-trained LLMs (GPT-3.5, Llama-3 70B, and Qwen-1.5 72B), when a model is shown an equal mixture of in-context samples across 3 tasks, the output distribution of the model is often a simplex over the correct outputs for each task, i.e. the top 3 next-token predictions for each input are the correct next-tokens for each of the 3 tasks, and different models exhibit biases towards different tasks. This is in contrast to the model either failing to learn the task to perform due to conflicting information in the context, or simply outputting the answer for one of the tasks that appears in the context.

Additionally, the authors show that, when training models to perform simple ICL tasks from scratch, e.g. simple arithmetic, the probability of each task’s output is proportional to the percentage at which that task’s examples appear in the context during test-time, even after the models are trained to perform ICL on a single-task at a time.

The authors then demonstrate theoretically, through the construction of a transformer model, that a transformer with 7 layers is capable of performing task superposition over a set of in-context examples.

Finally, the authors demonstrate that “task vectors”, defined as the model representation of the final token of the ICL prompt, reflects superposition. The authors show that the task vector created by a mixture of ICL tasks appears as an interpolation of each task’s task vector in T-SNE embeddings. Additionally, they show that creating new “task vectors” by interpolating the task vectors of distinct tasks can result in the model exhibiting task superposition as well.

The authors close by analyzing the relationship between task superposition and model size. They show that larger models have output probabilities that are more calibrated with the task probabilities of the context, i.e. the output for a task which appears in 50% of the in-context samples has a 50% probability in the model output.

**Strengths:**

The paper presents a relatively strong argument that LLMs can perform multiple tasks simultaneously when identifying the task from context. This, in turn, provides more evidence towards the notion that ICL operates as a “mixture of experts”, identifying a distribution of tasks from context and conditioning on that distribution when outputting an answer. As a result, this paper makes some important in-roads towards improving our understanding of ICL.

The “from-scratch” experiments are particularly strong in this setting, as they show a relatively tight relationship between the distribution of tasks in-context and the predicted distribution of the model over task outputs. Additionally, the analysis of how superposition occurs in task vectors is particularly convincing in demonstrating that there does indeed seem to be some kind of “task mixture” being identified by the model, and the analysis of how model scale impacts this seems to indicate that this task mixture capability becomes more accurate as model size increases, which is particularly useful in helping us understand how LLMs do ICL.

**Weaknesses:**

The “core” empirical results on large, pre-trained LLMs do not leave a clear take-away that superposition is occuring. This may, in part, be due to how the results are presented, i.e. the take-away that the model spreads it’s output across the 3 task outputs relies on the probabilities for “other” outputs to be low, which it is not always. This seems to only really be true for all models in one of the four settings considered. Thus, while the paper makes a strong case that LLMs can perform multiple tasks in superposition, it does not necessary make a strong case that current LLMs will perform multiple tasks in superposition.

Finally, the usefulness of task superposition is still up for debate - the authors note in their limitations section that it’s usefulness is limited due to the lack of decoding tools that can take advantage of it. However, there is also not an analysis of how task superposition can impact task performance, and that trade-off, if it exists, is not discussed in this work. It seems intuitive that task superposition may impact the performance of the tasks being superimposed, which may also have an effect on its usefulness. In fact, it is somewhat surprising to me that task performance is not a major consideration when making the claim that models can perform many tasks simultaneously.

**Questions:**

See weaknesses.

---

> ### Author Response · Authors · 2024-11-22
> **Response to Reviewer Ssa4 (1/1)**
>
> We thank the reviewer for their thoughtful feedback on our paper. We have updated our draft with a new Appendix E which contains additional experiments to address the reviewer's concerns. We break down the aforementioned weaknesses in two parts:
>
> **W1: Is superposition really happening on large pre-trained LLMs?**
>
> Yes. Based on your feedback, we conduct the following experiments that provide clear evidence that task superposition is occurring in current language models.
>
> In [Figure 8](https://github.com/76N5/superposition/blob/main/Figure_8.pdf) we show the probability of the most likely answer other than the task answer of tasks specified in the prompt. For example, in [Figure 8a](https://github.com/76N5/superposition/blob/main/Figure_8.pdf), we observe that for GPT-3.5 and Llama-3, the highest probability for a non-task answer (an individual “other” task) is still significantly lower than the four addition tasks that were given in the prompt. Because the next most likely answer still has lower probability than the specified task answers, any individual “other” answer must also has much lower probability than the specified tasks. This shows that the models focus on performing superposition on the tasks specified in-context. We note here that, Qwen-1.5 has similar probability as some of the task answers. We attribute this to the model’s limited ability to perform some of the specified tasks, specifically adding in language other than English.
>
> Our controlled ablation studies ([Figure 7](https://github.com/76N5/superposition/blob/main/Figure_7.pdf)) show that task probabilities directly reflect in-context demonstrations: task outputs have negligible probability when their corresponding task is absent from the prompt, but gain substantial probability when the task is included. This systematic behavior cannot be explained simply by noise.
>
> These findings demonstrate that task superposition is a real phenomenon that emerges specifically in response to multiple in-context tasks.
>
> (Experimental setup details can be found in Appendix E1 and E2 of our updated draft.)
>
>
> **W2: Analysis on how task superposition impact task performance and usefulness of task superposition**
>
> In [Table 2](https://github.com/76N5/superposition/blob/main/Table_2.pdf), we show the task accuracy with one task prompting $(K=1)$ and with multiple tasks prompting $(K>1)$, where we define correctness of a task as whether the correct task answer shows up in the top-$K$ output probabilities. We find that as we increase the number of tasks, all models exhibit an accuracy decrease in correctly performing each individual task. Notably, however, the accuracy degradation of Llama3-70B is less than that of Llama2-70B across most tasks. We believe this is indicative that improved model training techniques lead to better preservation of task superposition, and therefore, while task superposition can impact task performance, we believe such performance drop will be even lower for future models. Additionally, this also suggests that task superposition may be used to help benchmark future model training.
>
> (Experimental setup details can be found in Appendix E3 of our updated draft.)

---

> > ### Comment · Reviewer_Ssa4 · 2024-11-26
> >
> > Thank you for addressing my questions, in particular for including a discussion of how superposition affects task accuracy. It's interesting that some of these tasks do not seem to be significantly harmed by task superposition, whereas other task setting seem to suffer in a nearly catastrophic manner. Very low superposition performance seems to be somewhat correlated with task difficulty (how well the model performs the task in a single-task setting), which might suggest that this notion of superposition will break down for real-world tasks, as other reviewers have suggested. There is certainly some area for future work in this direction.
> >
> > I will keep my score the same, as I feel it sits somewhere between a 6 and 8: I feel the paper should be accepted, but I do still feel that it has some weaknesses such as limited practical take-aways with respect to applications of superposition and the new results still show that some models such as Qwen really struggle to do superposition but there is no substantial discussion of when and why some pre-trained models do or do not do task superposition.

---

> > > ### Author Response · Authors · 2024-11-27
> > >
> > > Thank you for your response! We will incorporate our discussions into our next revision.

---

### Author Response · Authors · 2024-11-22
**General Response by Authors**

# General Response

We sincerely appreciate the thoughtful and constructive feedback from the reviewers.

We include general update / clarification here in the general response. Other individual questions and comments from the reviewers are addressed separately in each reply.

(In some of our responses, we provide additional Figure / Table with anonymous link embedded for visualization. In case the link fails, Figure 7-8 & Table 2 can be found in Appendix E and Figure I-V can be found in Appendix F of our updated draft. We also fixed some small typos in our original submission. We will incorporate our discussion with the reviewers in our future revised manuscript and open source the code for running experiments.)

## A more explicit definition of task and task superposition
Here we define "task" formally in the setting of in-context learning.

Let $\mathcal{X}$ be the input space, $x\in\mathcal{X}$, and $\mathcal{Y}$ be the output space. We define a task as a function $g: \mathcal{X}\rightarrow \mathcal{Y}$; we denote a task answer as $g(x)$, and $(x, g(x))$ as a task example of the task $g$. Let $\texttt{prompt}=s_m\oplus x^{(m+1)}=[x^{(1)}, g(x^{(1)}),...,x^{(m)},g(x^{(m)}),x^{(m+1)}]$ be a prompt consisting of $m$ task examples and a query $x^{(m+1)}$. $M$ is a model that predicts $M(\texttt{prompt})\in\mathcal{Y}$. We say $M$ learns the task $g$ in-context if $M(\texttt{prompt})=g(x^{(m+1)})$, i.e., $M$ learns $g$ and performs the mapping on the query $x^{(m+1)}$.

Let $\texttt{LM}$ be a language model and let $\mathcal{V}$ be the vocabulary space (token space). Given a $\texttt{prompt}$, instead of outputing a $y\in\mathcal{Y}$ directly, $\texttt{LM}$ outputs a probability distribution over $\mathcal{Y}$, i.e., $\texttt{LM}(\texttt{prompt}) = [\mathbb{P}(y_1\mid \texttt{prompt}), ..., \mathbb{P}(y_{|\mathcal{Y}|}\mid \texttt{prompt})]$ for $y_j\in\mathcal{Y}, j=1,...,|\mathcal{Y}|$. (In practice $\texttt{LM}$ outputs a distribution over $\mathcal{V}$ but for simplicity here we assume inputs and outputs are single-tokens, i.e., $\mathcal{X}=\mathcal{V}=\mathcal{Y}$, and it naturally generalizes to the case when outputs are multi-tokens as we can predict the probability of multi-token output given prompt using a chain rule.) In previous works, that study in-context learning for language models [1,2], all task examples in a $\texttt{prompt}$ are from a **single** task $g$; $\texttt{LM}$ would put most of the probability mass on a single task answer, while the output is selected using decoding strategy like greedy decoding $M(\texttt{prompt})=\arg\max_{y}\mathbb{P}(y\mid \texttt{prompt})\in \mathcal{Y}$.

In our work, we are the first to study a setting that task examples in a single $\texttt{prompt}$ are from $K$ different tasks, i.e., we have tasks $g_1,...,g_{K}$ which form $\texttt{prompt}=[x^{(1)}, g^{(1)}(x^{(1)}),...,x^{(m)},g^{(m)}(x^{(m)}), x^{(m+1)}]$ where $g^{(j)}\in\{g_1,...,g_K\}$. Since $\texttt{LM}$ outputs a probability distribution over $\mathcal{Y}$, we calculate the probability mass it assigns to each of the task answers $g_1(x^{(m+1)}), ..., g_K(x^{(m+1)})$. In Figure 1 we depict $[\mathbb{P}(g_1(x^{(m+1)})\mid \texttt{prompt}),..., \mathbb{P}(g_K(x^{(m+1)})\mid \texttt{prompt})]$ and find that $\texttt{LM}$ assigns non-negligible probability on multiple task answers. In Section 3 we keep investigating this phenomenon and find that under various settings and for different models, given that the task examples in the $\texttt{prompt}$ are from multiple tasks, $\texttt{LM}$ will output non-negligible probabilities on the corresponding task answers. We term this phenomenon as *task superposition*.

We will incorporate this update in our revised manuscript.

[1] Brown, T. B. (2020). Language models are few-shot learners. arXiv preprint arXiv:2005.14165.

[2] Xie, S. M., Raghunathan, A., Liang, P., & Ma, T. (2021). An explanation of in-context learning as implicit bayesian inference. arXiv preprint arXiv:2111.02080.

---

### Comment · Area_Chair_pdQx · 2024-11-24
**Reminder: Author-Reviewer Discussion Period Closing Soon**

Dear Reviewers

This is a reminder that the author-reviewer discussion period will end on Nov 26 AoE.

Your engagement during this phase is critical for providing valuable feedback and clarifications. If you have any remaining questions or comments, please take a moment to participate before the deadline.

Thank you for your contributions to this important process.

AC

---

### Meta-Review · Area_Chair_pdQx · 2024-12-20

**Metareview:**

(a) Summary of Scientific Claims and Findings

This paper investigates a unique capability in Large Language Models (LLMs), referred to as “task superposition.” Through experiments, the authors demonstrate that LLMs can perform distinct in-context learning (ICL) tasks simultaneously within a single inference.

(b) Strengths of the Paper

The paper is the first to claim and explore the “task superposition” capability in LLMs. The authors empirically validate this phenomenon across various architectures and scenarios, showing its robustness.

(c) Weaknesses of the Paper and Missing Elements

The experiments primarily focus on synthetic or toy problems, with limited exploration of task superposition in real-world contexts.
The practical utility of task superposition in real-world applications requires further discussion and analysis.

(d) Decision and Rationale

While the paper introduces the intriguing concept of “task superposition,” it would benefit from further investigation into its implications for real-world tasks and performance. Addressing these gaps would significantly strengthen the contribution.

**Additional Comments On Reviewer Discussion:**

1. During the discussion phase, the authors provided additional clarification on the concept of task superposition and its novelty.

2. The paper received criticism for its limited focus on synthetic tasks rather than real-world applications.

3. Reviewer P2uz expressed skepticism about the novelty of task superposition, arguing that the described setting could be interpreted as a single complex task with outputs that are a mixture of several directions. This perspective is a valid critique.

---

### Decision · Program_Chairs · 2025-01-22

Reject